# Procurement Auctions via Approximately Optimal Submodular Optimization

Yuan Deng [1]   Amin Karbasi [2]   Vahab Mirrokni [1]   Renato Paes Leme [1]   Grigoris Velegkas [2] [*]   Song Zuo [1]

## Abstract

We study procurement auctions where an auctioneer must acquire services from strategic sellers with private costs. The quality of these services is represented by a known submodular function. Our goal is to design computationally efficient auctions that approximately maximize the difference between service quality and total seller costs, while remaining incentive compatible (IC), individually rational (IR), and yielding non-negative surplus (NAS) for the auctioneer. Our contribution is twofold: *(i)* we provide an improved analysis of existing algorithms for non-positive submodular function maximization; *(ii)* we develop frameworks that transform submodular optimization algorithms into mechanisms that are IC, IR, NAS, and preserve approximation guarantees. These frameworks apply both in an offline setting—where all bids are simultaneously observable—and an online setting—where sellers arrive adversarially and decisions must be made irrevocably. We further investigate whether state-of-the-art submodular algorithms can be converted into descending auctions under adversarially chosen price schedules. We show that any algorithm satisfying a bi-criteria $(\frac{1}{2}, 1)$-approximation in welfare can be adapted into such a descending auction. Finally, we demonstrate the practicality of our frameworks by instantiating them with various submodular optimization algorithms and evaluating their performance on datasets containing thousands of sellers.

---

[*]Part of the work was done while the author was a student researcher at Google Research. [1]Google Research [2]Yale University. Correspondence to: Yuan Deng <dengyuan@google.com>, Grigoris Velegkas <grigoris.velegkas@yale.edu>, Song Zuo <szuo@google.com >.

*Proceedings of the $42^{nd}$ International Conference on Machine Learning*, Vancouver, Canada. PMLR 267, 2025. Copyright 2025 by the author(s).

## 1. Introduction

In this paper, we consider procurement auctions (Dimitri et al., 2006) in which strategic service sellers with private costs submit bids to an auctioneer, who then decides the set of winners based on an objective function and purchases their services. Procurement auctions have been adopted in multitude of application domains, including industrial procurement (Bichler et al., 2006), data sharing (Roth & Schoenebeck, 2012; Rasouli & Jordan, 2021; Sim et al., 2022; Fallah et al., 2023), and crowdsourcing markets (Singer & Mittal, 2013). Additionally, Amazon Business provides government procurement solutions for products ranging from office supplies to first responder equipment (AmazonBusiness, 2024), while the U.S. Government Publishing Office conducts nationwide procurement for items used in publishing (USGPO, 2024). Each of these markets involve thousands of vendors for ensuring supplier diversity and competitive prices. There is also a large body of literature on procurement auction for the sale of data, including Bayesian mechanism design for one-sided markets, two-sided markets, and their online learning variants (Ghosh & Roth, 2011; Roth & Schoenebeck, 2012; Abernethy et al., 2015; Agarwal et al., 2019; Sim et al., 2022; Fallah et al., 2023; Agarwal et al., 2024).

Since procurement auctions were introduced to the algorithmic game theory community in the seminal paper of Nisan & Ronen (1999), many aspects of these auctions have been studied, including frugality in procurement auctions for minimizing purchasing costs (Archer & Tardos, 2007; Karlin & Kempe, 2005; Talwar, 2003), budget-feasible procurement auctions in which the purchasing cost is constrained by a budget (Singer, 2010; Kempe et al., 2010; Chen et al., 2011; Bei et al., 2012; Balkanski et al., 2022), and profit maximization for optimizing the auctioneer's surplus (Cary et al., 2008). In this paper, we consider the classic setting of procurement auctions with one auctioneer and a group of sellers, $\mathcal{N}$. Each seller has a *private* cost $c_i$ for providing the service, and for each $S \subseteq \mathcal{N}$, the auctioneer has a value function $f(S)$ for purchasing services from sellers in $S$. The social welfare obtained from the procurement auction when the auctioneer purchases from sellers in $S$ is given by the difference between the value obtained by the auctioneer and the total cost of sellers in $S$, i.e., $f(S) - \sum_{i \in S} c_i$.

We note that our objective differs from the utilitarian objective, which maximizes the sum of all agents' valuations, i.e., $f(S) + \sum_{i \notin S} c_i$, assuming seller $i$ has value $c_i$ if the service is not sold. Our objective can be considered a measure similar to the *gains-from-trade* in the bilateral trade literature, as it measures the *additional* total value generated by running the procurement auction. This is further motivated by the fact that, in many domains where procurement auctions can be applied, sellers incur costs only *after* they have been selected to provide a service. For instance, when a vendor provides goods or services to a company, it is most often the case that they incur costs only after signing the contract and needing to purchase the required materials. Although maximizing gains-from-trade is equivalent to maximizing the utilitarian objective, providing an approximation to the gains-from-trade is considerably harder than providing an approximation to the utilitarian objective. In particular, a non-zero approximation to the utilitarian objective may already be achieved without any trade, which would only result in a zero approximation to gains-from-trade.

Another major obstacle to studying our objective is that maximizing for $f(S) - \sum_{i \in S} c_i$ is a computationally challenging optimization problem, even when $f$ is a *monotone submodular* function. Submodularity captures a broad class of functions with diminishing returns, including gross-substitute functions, budget additive functions, and coverage functions. When $f$ is a monotone submodular function, the aforementioned optimization task is known as *regularized submodular maximization*, expressed as maximizing the difference between a monotone submodular function $f(S)$ and a modular function $\sum_{i \in S} c_i$ as the regularization term. For the special case where $c_i = 0$ for all sellers, it is well-known that obtaining an approximation ratio better than $(1 - 1/e)$ requires exponentially many queries to $f$ (Nemhauser et al., 1978). Before a breakthrough result from Sviridenko et al. (2017), many heuristics had been proposed in the literature (Feige et al., 2008; 2013; Kleinberg et al., 2004), but none of them provided provable guarantees that are universal and unconditional. For example, Feige et al. (2013) design algorithms with parameterized approximation guarantees, where the parameterization may depend on classes of instances and the properties of their optimal solutions. Sviridenko et al. (2017) show a tight *bi-criteria* $(\alpha, \beta)$-approximation guarantee of the form $f(S) - \sum_{i \in S} c_i \geq \alpha \cdot f(\mathsf{OPT}) - \beta \cdot \sum_{i \in \mathsf{OPT}} c_i$ with $\alpha = 1 - 1/e$ and $\beta = 1$, where $\mathsf{OPT} = \arg\max_S f(S) - \sum_{i \in S} c_i$, which rules out the possibility of constant approximation for the welfare objective. Harshaw et al. (2019) developed a simplified and practical algorithm, called *the distorted greedy algorithm*, that achieves the same optimal approximation guarantee.

Inspired by recent progress in regularized submodular maximization, we revisit the problem of procurement auctions where our objective is to maximize the welfare, i.e., the difference between the value obtained by the auctioneer and the total cost of sellers. Our goal is to design a mechanism that: **1)** is incentive compatible (IC) and individual rational (IR) for each seller; **2)** guarantees non-negative auctioneer surplus (NAS); **3)** achieves state-of-the-art bi-criteria welfare guarantees; **4)** is computationally efficient.

The interpretation of the desired game-theoretical properties, i.e., IC, IR, and NAS, is that IR encourages sellers to participate in the auction while IC prevents strategic behavior and simplifies the sellers' decision-making process. The surplus of the auctioneer is given by $f(S) - \sum_{i \in S} p_i$ for purchasing services from sellers in $S$, where $p_i$ is the payment to seller $i \in S$. NAS is reminiscent of the weakly budget-balanced property under the interpretation that $f(S)$ captures the potential revenue generated for the auctioneer through the services purchased from sellers in $S$. The NAS property is critical for many applications; if the auctioneer is at risk of having negative revenue, they might be incentivized not to run the auction at all. Thus, it is crucial to ensure that our transformations from algorithms to mechanisms satisfy NAS, which requires subtle technical work.

## 1.1. Our Results

In this paper, we make theoretical contributions to the literature of regularized submodular optimizations as well as theoretical and empirical contributions to the literature of procurement auctions.

On the submodular optimization side, Harshaw et al. (2019) show that the distorted greedy algorithm satisfies an $(1 - 1/e, 1)$ bi-criteria approximation guarantee. In Section 3, we demonstrate that the distorted greedy algorithm also satisfies $(1 - e^{-\beta}, \beta + o(1))$ bi-criteria approximation simultaneously for all $\beta \in [0, 1]$, which is also almost tight (Feldman, 2021). Using the framework developed in this work, we obtain mechanisms that satisfy the same approximation guarantees for all $\beta \in [0, 1]$. We also extend the results to a setting with noisy function evaluations.

Moving on to the mechanism design side in Section 4.1, from a theoretical perspective, we first show that VCG mechanisms satisfy IC, IR, and NAS, and they are always welfare-optimal, but it is computationally prohibitive to implement them for practical applications. We then develop a mechanism design framework that can convert state-of-the-art submodular optimization algorithms to sealed-bid mechanisms that satisfy IC, IR and NAS, preserve the bi-criteria welfare guarantees, and can be computed efficiently. Leveraging online submodular optimization algorithms, we extend our framework to the online setting where the sellers arrive in a potentially adversarial order and the auctioneer has to make an irrevocable decision whether to purchase their services.

In addition to sealed-bid mechanisms, in Section 5 we ask whether it is possible to convert submodular optimization algorithms to descending auctions. These auctions were initially designed under the assumption that $f$ is a gross-substitute function (Kelso Jr & Crawford, 1982), which is a subclass of submodular functions. We focus on the adversarial setting where the schedule of descending prices is determined by an adversary. We show that if the demand oracle is based on the cost-scaled greedy algorithm (Nikolakaki et al., 2021), the descending auction always achieves bi-criteria $(\frac{1}{2}, 1)$-approximation in welfare, even in the adversarial setting. On the other hand, we show that if the oracle solves the demand problem exactly, the approximation guarantees could be arbitrarily bad. We further establish a connection between descending auctions and online submodular optimization algorithms, showing that any online submodular optimization algorithm can be converted to a descending auction in an approximation-preserving way. Thus, an impossibility result, showing there is no descending auction that can achieve bi-criteria $(\alpha, 1)$-approximation in welfare with $\alpha > \frac{1}{2}$, implies an impossibility result on online submodular optimization with bi-criteria $(\alpha, 1)$-approximation guarantees for the same $\alpha$, which is a long-standing open question in online submodular optimization.

In Section 6, we complement our theoretical results with empirical studies evaluating the welfare performance and running time trade-offs of different mechanisms on a coverage problem. Due to space constraints, the discussions of further related work are deferred to Appendix A.

## 2. Preliminaries

We consider a setting of procurement auctions with one auctioneer and a set $\mathcal{N}$ of $n$ sellers with items to sell. The auctioneer has a valuation function $f : 2^{\mathcal{N}} \to \mathbb{R}_{\geq 0}$ that specifies the value that the auctioneer assigns to the items of every set $S$ of sellers, where $S \subseteq \mathcal{N}$. Each seller $i \in \mathcal{N}$ has a private cost $c_i \geq 0$ indicating the minimum acceptable payment for selling to the auctioneer. We focus on functions $f$ that are monotone and submodular with $f(\emptyset) = 0$. A function $f$ is monotone if $f(S) \leq f(T)$ for all $S \subseteq T \subseteq \mathcal{N}$, and submodular if it satisfies the property of diminishing returns: $f(i \mid S) >= f(i \mid T)$ for all $S \subseteq T \subset \mathcal{N}$ and $i \notin T$, where $f(i \mid S) = f(S \cup \{i\}) - f(S)$ computes the marginal contribution of seller $i$ to $f$, conditioned on $S$. Throughout the paper, we use bold symbols $\boldsymbol{x}$ to represent a vector with $n$ elements $(x_1, \cdots, x_n)$ and use $\boldsymbol{x}_{(a,b)}$ to represent $(x_a, \cdots, x_b)$. As usual, we use $-i$ to indicate all the sellers other than seller $i$.

Let $b_i$ be the reported bid from seller $i$. A mechanism $M = (a, p)$ consists of an allocation rule $a : \mathbb{R}_{\geq 0}^n \to 2^{\mathcal{N}}$ that maps sellers' reported bids $\boldsymbol{b}$ to a subset of sellers to procure the items, and a payment rule $p : \mathbb{R}_{\geq 0}^n \to \mathbb{R}_{\geq 0}^n$ that maps sellers' reported bids $\boldsymbol{b}$ to a vector of payments to each seller. We assume sellers have quasi-linear utilities such that given a bid profile $\boldsymbol{b}$ and a mechanism $M$, seller $i$'s utility is given by $u_i^M(\boldsymbol{b}) = p(\boldsymbol{b}) - c_i \cdot \mathbb{1}[i \in a(\boldsymbol{b})]$. Our goal is to design a mechanism $M$ that is incentive compatible, individual rational, and induces non-negative auctioneer surplus: A mechanism is incentive compatible (IC) if it is always an optimal strategy for a seller to report their private cost truthfully, i.e., for any seller $i$ and any $\boldsymbol{b}$, $u_i(c_i, \boldsymbol{b}_{-i}) \geq u_i(\boldsymbol{b})$; A mechanism is individual rational (IR) if a seller's utility is always non-negative if they report truthfully, i.e., for any seller $i$ and any $\boldsymbol{b}_{-i}$, $u_i(c_i, \boldsymbol{b}_{-i}) \geq 0$; A mechanism satisfies the non-negative auctioneer surplus (NAS) condition if the acquired value of the auctioneer is at least the total payment to the sellers, i.e., $f(a(\boldsymbol{b})) \geq \sum_{i \in 1}^n p_i(\boldsymbol{b})$, for any bid profile $\boldsymbol{b}$. We will refer to mechanisms that satisfy the IC, IR, and NAS conditions as *feasible* mechanisms. We measure the performance of a mechanism by its welfare $f(a(\boldsymbol{c})) - \sum_{i \in a(\boldsymbol{c})} c_i$ and let $\mathsf{OPT} = \arg\max_S f(S) - \sum_{i \in S} c_i$, when the definition of $f$ is clear from context. We may also write $c(S) = \sum_{i \in S} c_i$. We say that a mechanism satisfies bi-criteria $(\alpha, \beta)$-approximation to the welfare if $f(a(\boldsymbol{c})) - \sum_{i \in a(\boldsymbol{c})} c_i \geq \max\{0, \alpha \cdot f(\mathsf{OPT}) - \beta \cdot \sum_{i \in \mathsf{OPT}} c_i\}$.

## 3. Submodular Optimization Algorithms

We first present several submodular optimization algorithms that will be useful for the derivation of our mechanisms, and provide an improved analysis for the deterministic and stochastic versions of the distorted greedy algorithm (Harshaw et al., 2019). We will demonstrate how to convert all the algorithms from this section to NAS, IC, and IR mechanisms that maintain the approximation guarantees of the underlying algorithms in Section 4.1.

Recall that in the regularized submodular maximization problem under a cardinality constraint, there is a monotone submodular function $f : 2^{\mathcal{N}} \to \mathbb{R}_{\geq 0}$, a cost $c_i \in \mathbb{R}_{\geq 0}$ for each element $i \in \mathcal{N}$, and a cardinality constraint $k \leq n$. All the algorithms we touch upon in this section, and subsequently in our mechanism design framework, share the same paradigm (Algorithm 1): the algorithm maintains a candidate solution set initialized as $S = \emptyset$ and in each round $k$, it assigns a *score* to each element $i \in \mathcal{N} \setminus S$ based on a scoring function $G$, which depends on the cost vector $\boldsymbol{c}$ and possibly the round number $k$ as well as a random seed $r$ (for randomized algorithms). The algorithm then adds the element with the highest non-negative score to $S$.

We next describe each algorithm in detail, focusing on the corresponding scoring function $G$. The bi-criteria guarantees of all the algorithms are deferred to Table 1 in Appendix B. To simplify the notation, when we discuss deterministic algorithms, we omit referring to the random seed that $G$ could take as input, and we omit referring to the

round number $k$ as input of $G$ when $G$ does not use the round number information.

**Greedy-margin (Kleinberg et al., 2004).** We start with the simplest algorithm, called the greedy-margin algorithm, which is perhaps the most natural approach . This algorithm simply chooses the seller with the largest difference between their marginal contribution and their cost, i.e., the scoring function is given by $G(i, S, \boldsymbol{c}) = f(i \mid S) - c_i$.

**Greedy-rate (Feige et al., 2013).** The greedy-rate algorithm chooses the seller that maximizes the ratio of the difference between their marginal contribution and their cost, over their marginal contribution, i.e., the scoring function is given by $G(i, S, \boldsymbol{c}) = \frac{f(i|S)-c_i}{f(i|S)}$.

**Distorted Greedy (Harshaw et al., 2019).** The distorted greedy algorithm shares a similar flavor to the classical algorithm of Nemhauser et al. (1978), but with a slightly *distorted* objective with a multiplier $\left(1 - \frac{1}{n}\right)^{n-\ell}$ on the marginal contribution in round $\ell$: $G(i, S, \boldsymbol{c}, \ell) = \left(1 - \frac{1}{n}\right)^{n-\ell} \cdot f(i \mid S) - c_i$. It is worth highlighting that this scoring function does not have the diminishing-return structure and in particular, it does not stop early even if scores are negative for all remaining candidates.

**Stochastic Distorted Greedy (Harshaw et al., 2019).** In order to speed up the execution of the distorted greedy algorithm, Harshaw et al. (2019) proposed a randomized implementation of it that works as follows. It runs for $n$ iterations and in every iteration $k$ it draws a seller uniformly at random from $\mathcal{N}$. Assume the random seed $r$ encodes the selected seller in iteration $\ell$ via $r(\ell)$. Then, we can define the scoring function as $G(i, S, \boldsymbol{c}, k, \ell) = 1[i = r(\ell)] \cdot \left(\left(1 - \frac{1}{n}\right)^{n-\ell} \cdot f(i \mid S) - c_i\right)$. Harshaw et al. (2019) showed that, in expectation over the random draws of the sellers, the approximation guarantee of this algorithm does not degrade compared to its deterministic counterpart.

**Return-on-Investment (ROI) Greedy (Jin et al., 2021).** The ROI greedy algorithm chooses the seller that has the largest marginal contribution per unit of their cost among sellers whose cost not exceeding their marginal contribution, i.e., the scoring function is given by $G(i, S, \boldsymbol{c}) = \frac{f(i|S)-c_i}{c_i}$. Observe that, ROI greedy is effectively the same as greedy-rate as both algorithms are effectively ranking the sellers in descending order of $\frac{f(i|S)}{c_i}$. Feige et al. (2013) provide a parameterized approximation guarantee for this algorithm while Jin et al. (2021) demonstrate a unconditional approximation guarantee without paying a linear term on $f(\mathsf{OPT})$, which is desirable when this quantity is large.

**Cost-scaled Greedy (Nikolakaki et al., 2021).** This algorithm chooses the seller with the largest difference between their marginal contribution and *twice* their cost, i.e., the scoring function is given by $G(i, S, \boldsymbol{c}) = f(i \mid S) - 2 \cdot c_i$. In

fact, the cost-scaled greedy algorithm can also be applied to the online and adversarial setting in which the sellers arrive in an online manner (such that any decision is irrevocable) and the sequence of their arrival is determined by an adversary. In the online and adversarial setting, the algorithm maintains a tentative solution $S$ and adds a newly arrived seller to the solution $S$ if and only if $f(i \mid S) - 2 \cdot c_i > 0$.

### 3.1. Improved Analysis of Distorted Greedy

We now explain the improved analysis we propose for the distorted greedy algorithm. Recall that the distorted greedy score of every element $i \in \mathcal{N}$ in every round $1 \leq j \leq k$ of the execution of the algorithm is $\left(1 - \frac{1}{n}\right)^{k-j} \cdot f(i \mid S_{j-1}) - c_i$, and the element that maximizes it is added to the current solution, provided that its distorted score is non-negative. Harshaw et al. (2019) showed that both versions of the algorithm enjoy (roughly) a $(1 - 1/e, 1)$-bi-criteria approximation guarantee, which is tight. We show that, in fact, these algorithms satisfy an even stronger guarantee: the distorted greedy algorithm enjoys $(1 - e^{-\beta}, \beta + o(1))$-bi-criteria guarantee for all $\beta \in [0, 1]$, where the $o(1)$ term is sub-constant in cardinality $k$. A similar result holds, in expectation, for the stochastic version of the algorithm. The guarantee of the algorithm holds *simultaneously* for all $\beta \in [0, 1]$, so it does not require parameterization by $\beta$.

**Theorem 3.1.** *Let $\mathcal{N}$ be a universe of $n$ elements, $f : 2^{\mathcal{N}} \to \mathbb{R}_{\geq 0}$ be a monotone submodular function, and $c : \mathcal{N} \to \mathbb{R}_{\geq 0}$ be a cost function. Let $\mathsf{OPT}$ be the optimal solution of the objective $\max_{S \subseteq \mathcal{N}, |S| \leq k}\{f(S) - \sum_{i \in S} c_i\}$. Then, the output of the distorted greedy algorithm satisfies $f(R) - \sum_{j \in R} c_j \geq (1 - e^{-\beta})f(\mathsf{OPT}) - (\beta + 1/k)\sum_{j \in \mathsf{OPT}} c_j$, simultaneously for all $\beta \in [0, 1]$.*

The proof, as well as the formal statement for the the algorithm, are postponed to Appendix B. Our main technical insight is to perform a parameterized *analysis* of the potential function argument of Harshaw et al. (2019) based on the target value of $\beta$ that we wish to prove the guarantee for. In other words, given some $\beta \in [0, 1]$, we lower bound the potential function by a $\beta$-dependent quantity. This allows us to obtain the stated guarantees for all $\beta$ simultaneously. A result of Feldman (2021) (cf. Theorem B.1) shows that our analysis achieves the Pareto frontier of the bi-criteria guarantees for this problem, up to the $o(1)$ term. Details are deferred to Appendix B.

In Appendix B we present an adaptation of the distorted greedy algorithm that works even when we only have access to an approximate version $F : 2^{\mathcal{N}} \to \mathbb{R}_{\geq 0}$ of valuation function $f$ such that $(1-\varepsilon)f(S) \leq F(S) \leq (1+\varepsilon)f(S), \forall S \subseteq \mathcal{N}$ (Horel & Singer, 2016). Gong et al. (2023) propose a slight adaptation of the distorted greedy algorithm that performs well when $\varepsilon = O(1/k)$. However, when we convert their algorithm to a mechanism, it is not immediate how to

prove the NAS property, since $F$ might not be submodular, which was a crucial property of the function in our later proof of NAS. Thus, we propose a modification of their algorithm to overcome this issue (see Algorithm 7). Our main insight is to have the greedy scores of the elements in round $t$ of the execution depend not only on the current tentative solution $S_t$, but on the whole trajectory $S_1, \ldots, S_t$. Essentially, this enforces the structure of diminishing returns without hurting the approximation guarantees.

## 4. A Mechanism Design Framework

In this section, we develop a framework that is capable of converting the state-of-the-art submodular optimization algorithms to feasible mechanisms for procurement auctions. We first start with the "offline" setting, where all the sellers report their cost to the mechanism designer simultaneously and then we move on to the "online" setting, where the sellers arrive sequentially.

### 4.1. Offline Mechanism Design Framework

As a warm-up, we first show that the classic VCG framework (Vickrey, 1961; Clarke, 1971; Groves, 1973) provides mechanisms that are IC, IR, and welfare-efficient. It turns out that the VCG mechanisms also satisfy NAS.

**Proposition 4.1.** *The VCG mechanism satisfies NAS when $f$ is a submodular function.*

The proof is postponed to Appendix C. Although VCG mechanisms are IC, IR, NAS, and welfare-efficient, implementing them is computationally prohibitive. We now move on to describing the computationally efficient framework that transforms algorithms to mechanisms, which is one of the main contributions of our work.

Algorithm 1 provides a meta-algorithm $\mathcal{A} = (G)$ for regularized submodular function optimization specified by a scoring rule $G$, computing a score for a candidate $i$ given a subset $S$, a vector $\boldsymbol{c}$, the round number $k$, and possibly a random seed $r$ (for randomized algorithms) as input. The algorithm runs for $n$ rounds[1] and maintains a tentative solution set $S_k$ at the end of each round $k$. In each round $k$ it calls $G$ to compute a score for each candidate not in the tentative solution set $S_{k-1}$, and then it identifies the candidate $i^*$ with the highest score (where ties are broken lexicographically). If the highest score is positive, $S_k = S_{k-1} \cup \{i^*\}$; otherwise $S_k = S_{k-1}$.

**Assumption 4.2.** The meta-algorithm $\mathcal{A} = (G)$ satisfies: **1)** for all $i$ and $S$ with $i \notin S$, $G(i, S, \boldsymbol{b}, k, r)$ is non-increasing in $b_i$, for all $\boldsymbol{b}_{-i}$, $k$ and $r$; **2)** for all $i$ and $S$ with $i \notin S$,

---

**Algorithm 1** A meta algorithm $\mathcal{A} = (G)$ for submodular optimization

**Data:** A set of seller $\mathcal{N}$, a cost profile $\boldsymbol{c}$ from sellers, and a random seed $r$

**Result:** A subset of sellers to purchase services from

$S_0 = \emptyset$

**for** $k$ *from* $1$ *to* $n$ **do**

    $i^* = \arg\max_{i \notin S_{k-1}} G(i, S_{k-1}, \boldsymbol{c}, k, r)$

    **if** $G(i^*, S_{k-1}, \boldsymbol{c}, k, r) > 0$ **then**

        $\mid$   $S_k = S_{k-1} \cup \{i^*\}$

    **end**

    **else**

        $\mid$   $S_k = S_{k-1}$

    **end**

**end**

**return** $S_n$

---

if $b_i > f(i \mid S)$, then $G(i, S, \boldsymbol{b}, k, r) < 0$ for all $\boldsymbol{b}_{-i}$, $k$, and $r$; **3)** for all $i$, $S$ with $i \notin S$, $k$ and $r$, $G(i, S, \boldsymbol{b}, k, r)$ is independent of $\boldsymbol{b}_{-i}$.

We argue that both (1) and (2) of Assumption 4.2 are mild ones and any reasonable meta-algorithm $\mathcal{A}$ should satisfy it. For instance, all the algorithms we present in Table 1 satisfy these assumptions. In particular, Assumption 4.2(1) states that the scoring function $G$ should be non-increasing as the reported bid $b_i$ increases, which is a natural requirement as a candidate with a smaller reported bid is more favorable. Assumption 4.2(2) states that the algorithm should not pick a candidate whose marginal contribution is smaller than their reported bid. Under truthful reporting, such a candidate has a negative marginal contribution in round $k$ towards the social welfare, and therefore, they should not be included to the solution. Assumption 4.2(3) is a stronger assumption, stating that the score for a seller $i$ should be independent of bids from other sellers, but to the best of our knowledge, almost all state-of-the-art algorithms satisfy this assumption. In fact, for our mechanism to satisfy the desired properties we can relax Assumption 4.2(3) to require: for all $i$, $S$ with $i \in S$, $k$, and $r$, for any $\boldsymbol{b}$, if $i \neq \arg\max_{\ell \notin S} G(i, S, \boldsymbol{b}, k, r)$, then for any $b_i' > b_i$, $\arg\max_{\ell \notin S} G(\ell, S, (b_i', \boldsymbol{b}_{-i}), k, r) = \arg\max_{\ell \notin S} G(\ell, S, \boldsymbol{b}, k, r)$. In other words, as long as seller $i$ does not have the highest score, then the candidate with the highest score remains the same. Such a property is similar to *non-bossiness*, studied by Paes Leme et al. (2023).

Given a meta algorithm $\mathcal{A}$ specified by Algorithm 1, Algorithm 2 first runs Algorithm 1 with the reported bids as input in order to obtain the set of sellers $S^*$ whose items will be purchased. Then, the payment for each seller $i \in S^*$ is computed in the following way: we re-run $\mathcal{A}$ by raising the bid from seller $i$ to infinity and record the intermediate solutions $\{S_0, S_1, \cdots, S_n\}$. For

---

[1] Our framework can be extended to accommodate algorithms that stop early without running all the $n$ iterations. To simplify the exposition, we let the algorithm run for longer by adding extra *dummy* rounds.

each $S_k$, we compute the supremum of the set of bids $b_i \geq 0$ satisfying $i = \arg\max_{\ell \notin S_k} G(\ell, S_{k-1}, \boldsymbol{b}, k, r)$ and $G(i, S_{k-1}, \boldsymbol{b}, k, r) > 0$. If such a non-negative bid does not exist, the sup function takes a default value of 0. Finally, $p_i$ is computed by taking the max across $k \in [n]$.

---

**Algorithm 2** A feasible mechanism construction for a given meta algorithm $\mathcal{A}$

---

**Data:** A set of sellers $\mathcal{N}$, a bid profile $\boldsymbol{b}$ from sellers, and a meta algorithm $\mathcal{A}$

**Result:** A subset of sellers to purchase from and a vector of payment to sellers

Generate a random seed $r$ if needed or set $r = 0$
$S^* = \mathcal{A}(\mathcal{N}, \boldsymbol{b}, r)$
**for** $i \in S^*$ **do**
    Run $\mathcal{A}((\infty, \boldsymbol{b}_{-i}), r)$ and record $\{S_0, S_1, \cdots, S_n\}$
    $p_i = 0$
    **for** $k \in [n]$ **do**
        $p_i = \max(p_i, \sup\{b_i | G(i, S_{k-1}, \boldsymbol{b}, k, r) > 0\}$
        $p_i =$
        $\max(p_i, \sup\{b_i | i = \arg\max_{\ell \notin S_k} G(\ell, S_{k-1}, \boldsymbol{b}, k, r)\}$
    **end**
**end**
**return** $S^*$ *and* $\boldsymbol{p}$

---

**Theorem 4.3.** *For a meta-algorithm $\mathcal{A} = (G)$ satisfying Assumption 4.2, the mechanism constructed using Algorithm 2 is feasible.*

The proof is postponed to Appendix C. To show the IC and the IR properties, we make use of Myerson (1981) together with our carefully designed payment rule with Assumption 4.2(1) and 4.2(3). It is more technically difficult is to establish the NAS property, where we make use of the submodularity property of $f$ and Assumption 4.2(2). It is worth highlighting that Theorem 4.3 does not require the scoring function $G$ to have a diminishing-return structure, i.e., $G(i, S, \boldsymbol{b}, j, r) \geq G(i, T, \boldsymbol{b}, k, r)$ for all $S \subseteq T$ and $j \leq k$; and the distorted greedy algorithm does not satisfy such a structure. With Theorem 4.3, we establish a framework for converting a submodular optimization algorithm to a mechanism satisfying IC, IR, and NAS. In such a mechanism, the sellers are incentivized to submit their true private costs as their bids, and therefore, the mechanism preserves the bi-criteria welfare approximation guarantee, which follows immediately from the fact that under truthful bidding the allocations of Algorithm 1 and Algorithm 2 coincide.

*Remark* 4.4 (Running Time). In Algorithm 2, we need to make at most $O(n)$ many calls to the optimization algorithm in line to compute $\mathcal{A}((\infty, \boldsymbol{b}_{-i}), r)$, i.e., one call for each seller in the optimal solution. Moreover, for each inner loop starting, we need to make $O(n \log |B|)$ calls to the scoring function where $|B|$ is the number of possible bids. To summarize, Algorithm 2 makes $O(n)$ calls to the optimization

algorithm and $O(n^2 \log |B|)$ calls to the scoring function.

## 4.2. Online Mechanism Design Framework

In this section we shift our attention to the online setting, where each seller $i \in \mathcal{N}$ arrives online in an arbitrary order and the auctioneer needs to make an irrevocable decision of whether to buy the item or not. For convenience, assume seller $k$ arrives in round $k$. A meta-algorithm $\mathcal{A}^o = (G)$ for online submodular optimization is provided in Algorithm 3, where the scoring function $G$ in round $k$ computes a score for a seller $k$ given a subset $S$, a vector $\boldsymbol{c}_{(1,k)}$, and possibly a random seed $r$ as input. The algorithm maintains a tentative solution $S_k$ at the end of each round $k$. In each round $k$, seller $k$ is added to the tentative solution if and only if the scoring function returns a positive score. From the taxation principle, we can focus on (possibly randomized) *posted-price* mechanisms for designing IC and IR mechanisms, i.e., in each round $k$, the auctioneer makes a take-it-or-leave-it price $p_k \in \mathbb{R}_{\geq 0}$ for seller $k$. We provide an approximation-preserving transformation from online algorithms to posted-price mechanisms in Algorithm 8, postponed to Appendix C.1. Our main insight is to map the (online) scores to posted prices in an approximation preserving way. The formal proof is postponed to Appendix C.1. Similarly as in the offline setting, we require that the online submodular optimization algorithm satisfies the following mild assumptions.

**Assumption 4.5.** The meta-algorithm $\mathcal{A}^o = (G)$ satisfies: **1)** for all $k$, $S$, $\boldsymbol{b}_{(1,k-1)}$ and $r$, $G(k, S, \boldsymbol{b}_{(1,k)}, r)$ is continuous and strictly decreasing in $b_k$; **2)** for all $k$, $S$, $\boldsymbol{b}_{(1,k-1)}$ and $r$, if $b_k > f(k \mid S)$, then $G(k, S, \boldsymbol{b}_{(1,k)}, r) < 0$.

In particular, we drop the counterpart of Assumption 4.2(3) in the online setting, but we additionally require $G$ to be continuous and strictly decreasing to avoid tie-breaking issues so that the equation $G(k, S, (\boldsymbol{b}_{(1,k-1)}, z), r) = 0$ has a unique solution in terms of $z$.

**Theorem 4.6.** *For a meta-algorithm $\mathcal{A}^o = (G)$ satisfying Assumption 4.5, the online mechanism constructed using Algorithm 8 is IC, IR, NAS, and outputs the same solution as Algorithm 3 under full knowledge of the cost of the items.*

As we alluded to before in Section 3, the cost-scaled greedy algorithm (Nikolakaki et al., 2021) gives a $(\frac{1}{2}, 1)$-bi-criteria approximation guarantee in the online setting and fits within the template we have provided. Similarly, an adaptation of this algorithm by Wang et al. (2021) gives $(\beta-\alpha/\beta, \beta-\alpha/\alpha), 0 < \alpha \leq \beta$ parameterized bi-criteria guarantees in the online setting, and, interestingly, can also handle matroid constraints. In addition, Wang et al. (2020) provide parameterized bi-criteria guarantees adapting the algorithm from Nikolakaki et al. (2021). In the case of noisy evaluations of the function $f$, our framework can be applied

**Algorithm 3** A meta algorithm $\mathcal{A}^o = (G)$ for online submodular optimization

---

**Data:** A set of sellers arriving online and a random seed $r$
**Result:** A subset $S^*$ of sellers to purchase from
$S_0 = \emptyset, k = 0$
**while** *there exists a newly arrived seller $k + 1$ with cost $c_{k+1}$* **do**
    $k = k + 1$
    **if** $G(k, S_{k-1}, \boldsymbol{c}_{(1,k)}, r) > 0$ **then**
        $S_k = S_{k-1} \cup \{k\}$
    **end**
    **else**
        $S_k = S_{k-1}$
    **end**
**end**
**return** $S_k$

---

**Algorithm 4** Descending auction with a demand oracle $\mathcal{D}$ and a step size of $\varepsilon$

---

**Data:** A set of sellers $\mathcal{N}$ and a bid profile $\boldsymbol{b}$ from sellers
**Result:** A subset $S^*$ of sellers to purchase from and a vector of payment to sellers
Set the set of active sellers as $S = \mathcal{N}$
Set initial prices as $p_i = f(i \mid \emptyset)$
**while** $\mathcal{D}(S, \boldsymbol{p}) \subsetneq S$ **do**
    Select an arbitrary seller $i \in S \setminus \mathcal{D}(S, \boldsymbol{p})$
    $p_i = p_i - \varepsilon$
    **if** $p_i < b_i$ **then**
        $S = S \setminus \{i\}$
        $p_i = 0$
    **end**
**end**
**return** $S$ and $\boldsymbol{p}$

---

to the noise-robust algorithm from Gong et al. (2023), which builds upon Nikolakaki et al. (2021).

## 5. Descending Auctions

The mechanism design framework developed in Section 4.1 has a sealed-bid format. In this section, we investigate another popular class of procurement auctions, called *descending auctions*. A descending auction is parameterized by a demand oracle $\mathcal{D}$, that takes a set of sellers $S \subseteq \mathcal{N}$ and a vector of prices $\boldsymbol{p}$ as input and returns a subset of sellers $\mathcal{D}(S, \boldsymbol{p}) \subseteq S$. The auction maintains a set of active sellers $S$, initialized to be $\mathcal{N}$, and a vector of prices $\boldsymbol{p}$, where $p_i$ is initialized to be $f(i \mid \emptyset)$, the highest possible marginal contribution from seller $i$. In each iteration, the auction calls $\mathcal{D}$: If $\mathcal{D}(S, \boldsymbol{p}) = S$, the descending auction ends and returns the current $S$ as the set of sellers to purchase from and $\boldsymbol{p}$ as the vector of payment to sellers; If $\mathcal{D}(S, \boldsymbol{p}) \subsetneq S$, an active seller $i$ not in $\mathcal{D}(S, \boldsymbol{p})$ is chosen and its price $p_i$ is decreased by a small step size $\varepsilon$. If $p_i$ is smaller than $b_i$, seller $i$ is removed from the active seller set and $p_i$ is set to 0.

Algorithm 4 provides a classic paradigm of such auctions. Note that the descending auction is always IR since seller $i$ is chosen only if the final price $p_i$ is at least $b_i$, and IC since the reported bid $b_i$ is only used for checking whether the current price $p_i$ is smaller than $b_i$, which also enables the implementation without eliciting sealed bids from sellers in which whenever $p_i$ is lowered, the auctioneer asks the seller whether they would like to leave the market. This has the appealing property that for the set of the winning sellers, the auctioneer only learns an upper bound on their true valuation, instead of their actual valuation, which is always the case in sealed-bid auctions. Another advantage of descending auctions is that they satisfy *obviously strategyproofness* (Li, 2017). Moreover, NAS can be achieved if $f\big(\mathcal{D}(S, \boldsymbol{p})\big) - \sum_{i \in \mathcal{D}(S, \boldsymbol{p})} p_i \geq 0$ always holds.

Descending auctions were initially designed under the assumption that $f$ is a gross-substitute function, a subclass of submodular functions. In particular, when $f$ is a gross-substitute function, the demand oracle that exactly solves the welfare optimization problem can be computed in polynomial time; and with such an oracle, the descending auction returns the optimal subset OPT, even if it is executed in *the adversarial setting*, i.e., where the selection of seller $i$ not in $\mathcal{D}(S, p)$ for price decrement is determined by an adversary (Kelso Jr & Crawford, 1982; Paes Leme, 2017).

**Descending Auctions in the Adversarial Setting.** We first show that when $f$ is a submodular function, in the adversarial setting, a descending auction may return an arbitrarily worse solution even with the exact demand oracle satisfying bi-criteria $(1, 1)$-approximation in welfare. The proof is postponed to Appendix D.

**Theorem 5.1.** *In the adversarial setting, given demand oracle $\mathcal{D}$ satisfying bi-criteria $(1, 1)$-approximation in welfare, for any $L \in \mathbb{N}_+$, there exists a problem instance with $L + 2$ sellers and a vector of bids $\boldsymbol{b}$, such that Algorithm 4 with demand oracle $\mathcal{D}$ and $\varepsilon < \frac{1}{L}$ returns a subset $S^*$ with $f(S^*) - \sum_{i \in S^*} b_i = O\left(\frac{1}{L}\right) \cdot \big(f(\text{OPT}) - \sum_{i \in \text{OPT}} b_i\big)$.*

We complement our negative result by designing a demand oracle based on the cost-scaled greedy algorithm of (Nikolakaki et al., 2021), which leads to a descending auction with $(1/2, 1)$-approximation guarantees. The proof is postponed to Appendix D.

**Theorem 5.2.** *There exists a demand oracle $\hat{\mathcal{D}}$ satisfying bi-criteria $(\frac{1}{2}, 1)$-approximation in welfare such that, Algorithm 4 with demand oracle $\hat{\mathcal{D}}$ and $\varepsilon > 0$ always returns a subset $S^*$ satisfying $f(S^*) - \sum_{i \in S^*} b_i \geq \frac{1}{2} f(\text{OPT}) - \sum_{i \in \text{OPT}} b_i - n\varepsilon$ in the adversarial setting.*

*Remark* 5.3. We remark that our results show a, perhaps, counter-intuitive phenomenon in the adversarial setting:

there are instances in which if we run the descending auction described in Algorithm 4 with the *perfect* demand oracle we will end up with welfare that is arbitrarily worse than the execution with the oracle based on Nikolakaki et al. (2021). In particular, the family of problems that witnesses the lower bound in Theorem 5.2 shows that for every $L > 0$, there is an instance in which the solution $S^*$ that we get through the perfect oracle satisfies $f(S^*) - \sum_{i \in S^*} b_i < 1$, whereas the solution $\hat{S}$ that we get through the scaled-greedy oracle satisfies $f(\hat{S}) - \sum_{i \in \hat{S}} b_i \geq L/2 - 1$.

**From Online Submodular Optimization to Descending Auctions.** We demonstrate a reduction from online submodular optimization to descending auctions when the selection of seller $i$ not in $\mathcal{D}(S, p)$ for price decrement can be controlled by the auctioneer. Recall that we have demonstrated that an online submodular optimization algorithm (Algorithm 3) that satisfies Assumption 4.5 can be converted to an (online) posted-price auction that preserves the bi-criteria welfare guarantee. As posted-price auctions can be implemented by descending auctions with a tailored schedule of descending prices, an online submodular optimization algorithm (Algorithm 3) that satisfies Assumption 4.5 can also be converted to a descending auction that preserves the bi-criteria welfare guarantee (see Algorithm 9 in Appendix D). Thus, an impossibility result, that there is no descending auction that can achieve bi-criteria $(\alpha, 1)$-approximation in welfare with $\alpha > \frac{1}{2}$, directly implies an impossibility result on online submodular optimization with bi-criteria $(\alpha, 1)$-approximation guarantees for the same $\alpha$, which is a long-standing open question for online submodular optimization. We leave this as an interesting open question.

## 6. Experiments

In this section, we empirically evaluate the welfare performance and running time trade-offs of various mechanisms on a publicly available coverage problem. Our instances are constructed using a bipartite graph from SNAP (https://snap.stanford.edu/data/wiki-Vote.html). The graph consists of approximately 7000 nodes representing *sets* and 2800 nodes representing *vertices* to be covered by the sets. We consider the value of covering each vertex and the cost of selecting each set based on the degrees of the corresponding nodes. We define the value of $f(S)$ as the sum of the values from vertices covered by at least one selected set, i.e., $f(S) = \sum_{i \in \bigcup_{s \in S} s} v_i$. To create instances with varying sizes and value-to-cost ratios, for each pair $(n, s)$, we randomly sample subsets of the sets of size $n$ and scale their costs using $\kappa \sim U[s, s^2]$. We generate 10000 random instances per $(n, s)$. Since VCG needs to solve exactly $\arg\max_S \left( f(S) - \sum_{i \in S} c_i \right)$, we solve the optimization

problem using a mixed integer program (MIP) when $n$ is relatively small. Figure 1 shows the running time comparison of different methods and Figure 2 compares the welfare outcomes. In summary, **i)** VCG computes both VCG allocation and payment; **ii)** Optimal Welfare computes VCG allocation (i.e., the welfare-optimal allocation) only; **ii)** For Descending Auction (DA) with an unspecified oracle, we implement it with a demand oracle that computes the welfare-optimal allocation. We also experiment with Greedy-margin, Greedy-rate, Distorted Greedy, and Cost-scaled Greedy, introduced in Section 3 and their DA variants. In Appendix E, we use an approach, based on lazy implementations of the greedy algorithm (Minoux, 2005), to speed up the computation for both allocation and payment.

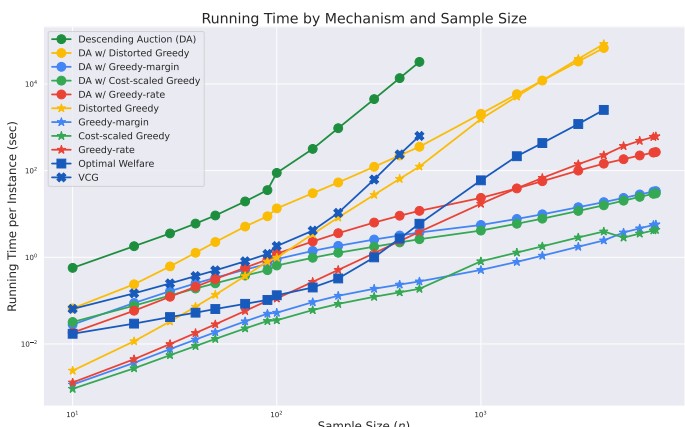

*Figure 1.* Average running time for different mechanisms at different sample sizes $n$. Both axes are log-scaled.

**Running Time Comparison** Figure 1 shows the average running time for different mechanisms. VCG, Descending Auction, and Optimal Welfare exhibit super-polynomial complexity due to MIP computations, with Descending Auction being the slowest. Greedy-based algorithms demonstrate polynomial complexity, but Distorted Greedy's time grows faster due to the absence of a diminishing-return structure, so the approach from Appendix E does not apply.

**Welfare Comparison** Figure 2 compares welfare for sample sizes $n = 4000$. Further comparisons are shown in Figure 3 and 4, Appendix E. Instances are grouped by the fraction of active agents, determined by whether their marginal contribution exceeds their cost. This fraction decreases as the cost multiplier $\kappa$ increases. When feasible to run, DA with an optimal oracle outperforms other auctions—contrasting our theoretical result that DA can be much worse than cost-scaled greedy in some cases. Across approximation algorithms, direct implementations outperform their DA variants, with an ordering of Greedy-margin > Greedy-rate > Cost-scaled Greedy > Distorted Greedy.

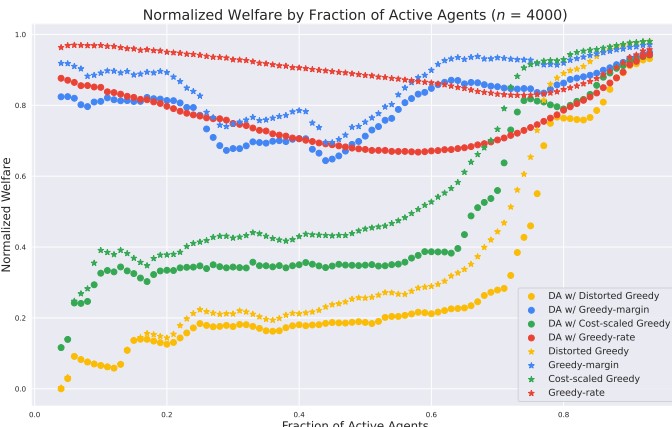

*Figure 2.* Welfare as a function of the fraction of active agents for $n = 4000$. Curves correspond to different mechanisms.

## 7. Conclusion

In this work we propose a new procurement auction setting inspired by the recent development in regularized submodular optimizations. Our results enable computational and welfare efficient transformations from regularized submodular maximization algorithms to various types of mechanisms, including sealed-bid auctions and descending auctions, that satisfy several desirable properties such as NAS, IC, and IR. Moreover, we have tested our framework on several large-scale instances which showcases its practical applicability. An immediate direction is to close the gap between the $(1/2, 1)$ bound of our descending auction and the $(1 - 1/e, 1)$ we get in the sealed-bid auction. Another interesting problem would be to replace the cost in the objective of the mechanism designer with the payment, i.e., to study an objective of maximizing the surplus of the mechanism designer. It would also be interesting to see if the descending auction can get the same performance as VCG, when we disregard computational considerations.

## Acknowledgements

Amin Karbasi acknowledges funding in direct support of this work from NSF (IIS-1845032), ONR (N00014- 19-1-2406), and the AI Institute for Learning-Enabled Optimization at Scale (TILOS). Grigoris Velegkas was supported in part by the AI Institute for Learning-Enabled Optimization at Scale (TILOS).

## Impact Statement

This paper presents work whose goal is to advance the field of Machine Learning. There are many potential societal consequences of our work, none which we feel must be specifically highlighted here.

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

# A. Related Work

**Gains From Trades vs. Utilitarian Objectives.** The objective we study in this work has connections to gains-from-trade in two-sided market. The seminal work of Myerson & Satterthwaite (1983) showed that even when there is one seller and one buyer there is no mechanism that satisfies IR, Bayesian IC (BIC), Budget-Balance, and can extract the full gains-from-trade (GFT), i.e., the total value generated by transferring the items from the sellers to the buyers. In light of that result, a lot of works have focused on relaxing one of these desiderata. Of particular interest are the works that are seeking an *approximately* optimal solution to the GFT objective. In the case of one buyer and one seller, McAfee (1992) provided an elegant mechanism that achieves an $1/2$-approximation guarantee if the median of the buyer's value is higher than that of the seller's. In a similar spirit, Blumrosen & Mizrahi (2016) proposed a $1/e-$approximation for this problem, when both the buyer and the seller satisfy the monotone-hazard-rate (MHR) condition. Recently, Brustle et al. (2017) provided a simple mechanism that gives a $1/2-$approximation to GFT under arbitrary distributions. Subsequently, Deng et al. (2022) provided a constant approximation to the *first-best* objective, i.e., the GFT that can be achieved if the agents are not strategic, resolving a long-standing open question; and Fei (2022) improves the approximation factor to 3.15. Cai et al. (2021) moved beyond the setting with one seller, providing approximations algorithms in environments with multiple sellers. A different line of work in the two-sided markets literature has focused on the utilitarian objective, i.e., the total value of the buyers for the items that the purchased and the total cost of the sellers who did not sell their items. Lehmann et al. (2001) gave a greedy algorithm for this objective that gives a $1/2-$approximation, and Fu et al. (2012) implemented it as an ascending auction that gives the same approximation, when the buyers are not strategic. More recently, Blumrosen & Dobzinski (2021) designed a simple take-it-or-leave-it mechanism for this objective that achieves a $(1-1/e)-$approximation. Colini-Baldeschi et al. (2016) designed a constant-factor approximation mechanism that satisfies the Strongly Budget Balanced condition, meaning that all the payments of the buyers are transferred to the sellers. Subsequently, Colini-Baldeschi et al. (2020) developed a constant-factor approximation mechanism in the setting with XOS valuations.

**Budget-feasible Mechanism Design.** The study of budget-feasible mechanism design, with an emphasis on procurement auctions, was put forth by the seminal work on Singer (2010) who provided a prior-free[2] budget-feasible mechanism that enjoys a constant approximation when the objective is a *non-negative* and monotone submodular function. Since then, a long line of work has studied this problem providing better approximation guarantees, relaxing the monotonicity constraint on the objective function and extending the results to even more general classes of functions including XOS and subadditive functions (Dobzinski et al., 2011; Chen et al., 2011; Anari et al., 2014; Bei et al., 2017; Jalaly Khalilabadi & Tardos, 2018; Amanatidis et al., 2019; Balkanski et al., 2022; Han et al., 2024). Currently, the best approximation guarantees for monotone submodular functions are obtained by Han et al. (2024), that achieve an approximation factor of 4.3. It is worth highlighting that, to the best of our knowledge, none of these works have studied *non-positive* submodular functions.

**Reduction from Algorithm Design to Mechanism Design.** Our results contribute to the rich literature of transformations from *algorithms* that operate on non-strategic data to *mechanisms* whose input is coming from strategic agents. A striking result by Chawla et al. (2012) showed that there is no welfare-preserving black-box reduction when the mechanism is required to be truthful in expectation. Since then, a beautiful line of work initiated by Hartline & Lucier (2010) develops mechanisms that satisfy the weaker Bayesian Incentive Compatibility (BIC) condition, instead of truthfulness in expectation. An important tool that all these works (Hartline & Lucier, 2010; Bei & Huang, 2011; Hartline et al., 2011; 2015; Dughmi et al., 2017) utilize to establish the (approximate) BIC property is a *replica-to-surrogate* matching. It is worth highlighting that Dughmi et al. (2017) managed to obtain an *exactly* BIC mechanism, by introducing novel constructions in the context of mechanism design, such as various *Bernoulli factories*. In the context of multidimensional revenue maximization, Cai et al. (2012b;a; 2013a;b) developed black-box transformations from algorithms to (approximately) revenue-optimal and (approximately) BIC mechanisms

**Submodular Optimization.** The problem of submodular maximization has received a lot of attention in the optimization literature. The seminal work of Nemhauser et al. (1978) shows that the natural greedy algorithm gives an $(1-1/e)-$approximation to the optimal solution when the submodular function is *monotone* and *non-negative*. Later, Feige et al. (2011) designed algorithms that provide constant approximation guarantees when the underlying function is *non-negative*, but, potentially, non-monotone. Designing approximation algorithms with constant-factor approximation guarantees for general non-positive submodular functions is known to be hard (Papadimitriou & Yannakakis, 1988; Feige, 1998). The study of the objective $f(S) - c(S)$, where $f$ is a non-negative submodular function and $c$ is a linear function, is

---

[2]This means that the auctioneer does not have any prior information about the private types of the sellers.

commonly referred to as *regularized submodular optimization*. This line of work was initiated by Sviridenko et al. (2017) who designed an algorithm that obtains a $(1 - 1/e)f(\text{OPT}) - c(\text{OPT})$ and showed that this guarantee is optimal. Later, Feldman (2021) and Harshaw et al. (2019) designed simpler and more computationally efficient algorithms that attain the same approximation guarantees. Since then, there has been a long line of work studying that problem and designing algorithms that work in the centralized setting, the distributed setting, the streaming setting, and the online setting (Kazemi et al., 2021; Wang et al., 2020; 2021; Jin et al., 2021; Mitra et al., 2021; Nikolakaki et al., 2021; Gong et al., 2021; Tang & Yuan, 2021a;b; Geng et al., 2022; Lu et al., 2023a; Qi, 2023; Lu et al., 2023b; Gong et al., 2023). It is worth noting that the design of approximation algorithms for bi-criteria objectives has alson been considered in different contexts in the past (Kleinberg et al., 2004; Feige et al., 2008; 2013).

# B. Omitted Details from Section 3

*Table 1.* Instantiations of Algorithm 2 with different submodular optimization algorithms.

| Algorithm | Approximation Guarantee |
|---|---|
| Greedy-margin (Kleinberg et al., 2004) | No Worst-Case Guarantee |
| Greedy-rate (Feige et al., 2013) | Parametrized Guarantee |
| Distorted Greedy (Harshaw et al., 2019) & (this paper) | $(1 - e^{-\beta}) \cdot f(\text{OPT}) - (\beta + o(1)) \cdot c(\text{OPT})$ for all $\beta \in [0, 1]$ |
| Stochastic Distorted Greedy (Harshaw et al., 2019) & (this paper) | $(1 - e^{-\beta}) \cdot f(\text{OPT}) - (\beta + o(1)) \cdot c(\text{OPT})$ for all $\beta \in [0, 1]$ (in expectation) |
| ROI Greedy (same as Greedy-rate) (Jin et al., 2021) | $f(\text{OPT}) - \left(1 + \ln \frac{f(\text{OPT})}{c(\text{OPT})}\right) \cdot c(\text{OPT})$ |
| Cost-scaled Greedy (Nikolakaki et al., 2021) | $1/2 \cdot f(\text{OPT}) - c(\text{OPT})$ (online and adversarial) |

## B.1. Deterministic Distorted Greedy Algorithm Analysis

---
**Algorithm 5** Distorted greedy algorithm (Harshaw et al., 2019)
---
**Data:** A set of $n$ items $\mathcal{N}$, a monotone submodular function $f : 2^{\mathcal{N}} \to \mathbb{R}_{\geq 0}$, a cost function $c : \mathcal{N} \to \mathbb{R}_{\geq 0}$, a capacity constraint $k \leq n$

**Result:** A subset of the items $R \subseteq \mathcal{N}$ with $|S| \leq k$

$S_0 = \emptyset$ **for** $j$ *from* $1$ *to* $k$ **do**

   $G(i, S_{j-1}, \boldsymbol{c}, j, r) = \left(1 - \frac{1}{n}\right)^{k-j} \cdot f(i \mid S_{j-1}) - c_i, \forall i \in S$   $i_t = \arg\max_{i \notin S_{j-1}} G(i, S_{j-1}, \boldsymbol{c}, j, r)$ **if**
   $G(i^*, S_{j-1}, \boldsymbol{c}, j, r) > 0$ **then**

     | $S_j = S_{j-1} \cup \{i^*\}$

   **end**

   **else**

     | $S_j = S_{j-1}$

   **end**

**end**

**return** $S_n$

---

In this section we extend the results of (Harshaw et al., 2019) that obtain a $(1 - e^{-1}, 1)$ bi-criteria approximation guarantee to $(1 - e^{-\beta}, \beta)$ guarantee, where $\beta \in [0, 1]$. First, we recall a negative result from (Feldman, 2021).

**Theorem B.1** ((Feldman, 2021)). *For every* $\beta \in [0, 1], \varepsilon > 0$, *no polynomial-time algorithm can guarantee* $(1 - e^{-\beta} + \varepsilon, \beta)$- *bi-criteria approximation for the problem* $\max_S (f(S) - \sum_{i \in S} c_i)$.

We underline that Feldman (2021) proposes an algorithm that achieves an (almost) matching upper bound, however the algorithm works using the continuous multilinear extension of the underlying function, which is quite non-practical and does not fit our mechanism design framework. We propose an approach based on the distorted greedy algorithm of (Harshaw et al., 2019) that achieves this guarantee for every $\beta \in [0, 1]$. Formally, we prove the result stated in Theorem 3.1.

We reiterate that Theorem 3.1 improves upon the guarantees stated in (Harshaw et al., 2019) since it achieves optimal bi-criteria guarantee simultaneously for all $\beta \in [0, 1]$. Following (Harshaw et al., 2019), we define the following functions

$$\Phi_t(T) = \left(1 - \frac{1}{k}\right)^{k-t} f(T) - \sum_{j \in T} c_j, \qquad\qquad \forall T \subseteq \mathcal{N},$$

$$\Psi_t(T, i) = \max\left\{0, \left(1 - \frac{1}{k}\right)^{k-(t+1)} f(i|T) - c_i\right\}, \qquad\qquad \forall T \subseteq \mathcal{N}, i \in \mathcal{N}.$$

The next result from (Harshaw et al., 2019) describes the connection between these two quantities.

**Lemma B.2** ((Harshaw et al., 2019)). *In each iteration* $t \in \{0, \ldots, k-1\}$ *of Algorithm 5 it holds that*

$$\Phi_{t+1}(S_{t+1}) - \Phi_t(S_t) = \Psi_t(S_t, i_t) + \frac{1}{k}\left(1 - \frac{1}{k}\right)^{k-(t+1)} f(S_t).$$

The next result of (Harshaw et al., 2019) relates the marginal gain in each iteration of Algorithm 5 to the function $\Psi_t(S_t, i_t)$.

**Lemma B.3** ((Harshaw et al., 2019)). *In each iteration of Algorithm 5 it holds that*

$$\Psi_t(S_t, i_t) \geq \frac{1}{k}\left(1 - \frac{1}{k}\right)^{k-(t+1)} (f(\mathsf{OPT}) - f(S_t)) - \frac{1}{k}c(\mathsf{OPT})$$

Finally, we will make use of the following simple technical result.

**Proposition B.4.** *For any* $n \in \mathbb{N}, \beta \in [0, 1]$ *it holds that*

$$\frac{1}{n} \cdot \frac{1 - \left(1 - \frac{1}{n}\right)^{x \cdot n}}{1 - \left(1 - \frac{1}{n}\right)} \geq 1 - e^{-x}.$$

*Proof.* We have that

$$\frac{1}{n} \cdot \frac{1 - \left(1 - \frac{1}{n}\right)^{x \cdot n}}{1 - \left(1 - \frac{1}{n}\right)} = 1 - \left(1 - \frac{1}{n}\right)^{x \cdot n} \qquad\qquad \text{(Simplify denominator)}$$

$$\geq 1 - e^{-\frac{x \cdot n}{n}}. \qquad\qquad (1 - x \leq e^{-x})$$

$\square$

Equipped with the previous results, we are ready to prove Theorem 3.1.

*Proof of Theorem 3.1.* By definition of $\Phi_0(\cdot), \Phi_k(\cdot)$ we have that

$$\Phi_0(S_0) = \left(1 - \frac{1}{k}\right) \cdot f(\emptyset) - c(\emptyset) = 0,$$

and

$$\Phi_k(S_k) = \left(1 - \frac{1}{k}\right)^0 \cdot f(S_K) - c(S_k) = f(S_k) - c(S_k).$$

Notice that $\Phi_k(S_k) - \Phi_0(S_0) = \sum_{t=0}^{k-1} \Phi_{t+1}(S_{t+1}) - \Phi_t(S_t)$. By Lemma B.2 it follows immediately that $\Phi_{t+1}(S_{t+1}) - \Phi_t(S_t) \geq 0$. Moreover, by Lemma B.2 and Lemma B.3 we have that

$$\Phi_{t+1}(S_{t+1}) - \Phi_t(S_t) = \Psi_t(S_t, i_t) + \frac{1}{k}\left(1 - \frac{1}{k}\right)^{k-(t+1)} f(S_t) \qquad\qquad \text{(Lemma B.2)}$$

$$\geq \frac{1}{k}\left(1-\frac{1}{k}\right)^{k-(t+1)}(f(\mathsf{OPT})-f(S_t))$$

$$-\frac{1}{k}c(\mathsf{OPT})+\frac{1}{k}\left(1-\frac{1}{k}\right)^{k-(t+1)}f(S_t) \qquad \text{(Lemma B.3)}$$

$$\geq \frac{1}{k}\left(1-\frac{1}{k}\right)^{k-(t+1)}f(\mathsf{OPT})-\frac{1}{k}c(\mathsf{OPT}) \qquad \text{(Rearranging terms)}$$

First, assume that $\beta$ is a multiple of $1/k$. We lower bound the first $k - \beta \cdot k$ terms of $\sum_{t=0}^{k-1}\Phi_{t+1}(S_{t+1}) - \Phi_t(S_t)$ by 0 and the last $\beta \cdot k$ terms by the previous inequality. Thus, we get

$$\sum_{t=0}^{k-1}\Phi_{t+1}(S_{t+1}) - \Phi_t(S_t) \geq \sum_{t=k-\beta\cdot k}^{k-1}\Phi_{t+1}(S_{t+1}) - \Phi_t(S_t)$$

$$\geq \sum_{t=k-\beta\cdot k}^{k-1}\left\{\frac{1}{k}\left(1-\frac{1}{k}\right)^{k-(t+1)}f(\mathsf{OPT})-\frac{1}{k}c(\mathsf{OPT})\right\}$$

$$= \left(\sum_{t=k-\beta\cdot k}^{k-1}\frac{1}{k}\left(1-\frac{1}{k}\right)^{k-(t+1)}\right)f(\mathsf{OPT})-\beta\cdot c(\mathsf{OPT})$$

$$= \frac{1}{k}\cdot\frac{1-\left(1-\frac{1}{k}\right)^{\beta\cdot k}}{1-\left(1-\frac{1}{k}\right)}\cdot f(\mathsf{OPT})-\beta\cdot c(\mathsf{OPT}) \qquad \text{(Sum of geometric series)}$$

$$\geq (1-e^{-\beta})\cdot f(\mathsf{OPT})-\beta\cdot c(\mathsf{OPT}). \qquad \text{(Proposition B.4)}$$

In case $\beta$ is not a multiple of $1/k$, the same analysis goes through with $\hat{\beta}$ being the smallest multiple of $1/k$ that is greater than $\beta$ and the guarantee we get is $(1-e^{-\hat{\beta}},\hat{\beta})$, which is at least $(1-e^{-\beta},\beta+1/k)$. □

### B.2. Stochastic Distorted Greedy Algorithm Analysis

In this section we shift our attention to the stochastic version of the distorted greedy algorithm (Harshaw et al., 2019), which requires fewer oracle calls to the function $f(\cdot)$ than its deterministic counterpart.

---

**Algorithm 6** Stochastic distorted greedy algorithm (Harshaw et al., 2019)

---

**Data:** A set of $n$ items $\mathcal{N}$, a monotone submodular function $f : 2^{\mathcal{N}} \to \mathbb{R}_{\geq 0}$, a cost function $c : \mathcal{N} \to \mathbb{R}_{\geq 0}$, a capacity constraint $k \leq n$, an error parameter $\varepsilon > 0$
**Result:** A subset of the items $R \subseteq \mathcal{N}$ with $|S| \leq k$
$S_0 = \emptyset$ $\ s = \lceil \frac{n}{k}\log\frac{1}{\varepsilon}\rceil$ **for** $j$ *from* $1$ *to* $k$ **do**

  $B_j =$ sample $s$ elements uniformly and independently from $\Omega$ $\ G(i, S_{j-1}, \boldsymbol{c}, j, r) = \left(1-\frac{1}{n}\right)^{k-j}\cdot f(i \mid S_{j-1})-c_i, \forall i \in$
  $B_j$ $\ i_t = \arg\max_{i\notin S_{j-1}, i\in B_j} G(i, S_{j-1}, \boldsymbol{c}, j, r)$ **if** $G(i^*, S_{j-1}, \boldsymbol{c}, j, r) > 0$ **then**
  | $\ S_j = S_{j-1}\cup\{i^*\}$
  **end**
  **else**
  | $\ S_j = S_{j-1}$
  **end**
**end**
**return** $S_n$

---

We first state some results from Harshaw et al. (2019) that will be useful in our analysis.

**Lemma B.5** ((Harshaw et al., 2019)). *In each step $t$ of Algorithm 6 it holds that*

$$\mathbb{E}[\Psi_t(S_t, i_t)] \geq (1-\varepsilon)\left(\frac{1}{k}\left(1-\frac{1}{k}\right)^{k-(t+1)}(f(\mathsf{OPT}) - \mathbb{E}[f(S_t)]) - \frac{1}{k}c(\mathsf{OPT})\right)$$

Notice also that we have the trivial lower bound $\mathbb{E}[\Psi_t(S_t, i_t)] \geq 0$. We are now ready to state and prove our main result in this subsection.

**Theorem B.6.** *Let $\mathcal{N}$ be a universe of $n$ elements, $f : 2^{\mathcal{N}} \to \mathbb{R}_{\geq 0}$ be a monotone submodular function and $c : \mathcal{N} \to \mathbb{R}_{\geq 0}$ be a cost function. Let $\mathsf{OPT}$ be the optimal solution of the objective $\max_{S \subseteq \mathcal{N}, |S| \leq k}\{f(S) - \sum_{i \in S} c_i\}$. Then, the output $R$ of Algorithm 6 satisfies $\mathbb{E}\left[f(R) - \sum_{j \in R} c_j\right] \geq (1-\varepsilon)(1-e^{-\beta})f(\mathsf{OPT}) - (\beta + 1/k)\sum_{j \in \mathsf{OPT}} c_j$, simultaneously for all $\beta \in [0, 1]$.*

*Proof.* By definition of $\Phi_0(\cdot), \Phi_k(\cdot)$ we have that

$$\mathbb{E}[\Phi_0(S_0)] = \left(1-\frac{1}{k}\right) \cdot f(\emptyset) - c(\emptyset) = 0\,,$$

and

$$\mathbb{E}[\Phi_k(S_k)] = \mathbb{E}\left[\left(1-\frac{1}{k}\right)^0 \cdot f(S_K) - c(S_k)\right] = \mathbb{E}\left[f(S_k) - c(S_k)\right]\,.$$

Notice that, by linearity of expectation, $\mathbb{E}[\Phi_k(S_k) - \Phi_0(S_0)] = \sum_{t=0}^{k-1}\mathbb{E}[\Phi_{t+1}(S_{t+1}) - \Phi_t(S_t)]$. By definition of $\Psi_{t+1}$ it follows immediately that $\mathbb{E}[\Phi_{t+1}(S_{t+1}) - \Phi_t(S_t)] \geq 0$.[3] Moreover, by Lemma B.2 and Lemma B.5 we have that

$$\mathbb{E}[\Phi_{t+1}(S_{t+1}) - \Phi_t(S_t)] = \mathbb{E}\left[\Psi_t(S_t, i_t) + \frac{1}{k}\left(1-\frac{1}{k}\right)^{k-(t+1)}f(S_t)\right] \quad \text{(Lemma B.2 and linearity of expectation)}$$

$$\geq (1-\varepsilon)\left(\frac{1}{k}\left(1-\frac{1}{k}\right)^{k-(t+1)}(f(\mathsf{OPT}) - \mathbb{E}[f(S_t)]) - \frac{1}{k}c(\mathsf{OPT})\right) +$$

$$\frac{1}{k}\left(1-\frac{1}{k}\right)^{k-(t+1)}\mathbb{E}[f(S_t)] \quad \text{(Lemma B.3)}$$

$$\geq (1-\varepsilon)\left(\frac{1}{k}\left(1-\frac{1}{k}\right)^{k-(t+1)}f(\mathsf{OPT}) - \frac{1}{k}c(\mathsf{OPT})\right) +$$

$$\frac{\varepsilon}{k}\left(1-\frac{1}{k}\right)^{k-(t+1)}\mathbb{E}[f(S_t)] \quad \text{(Rearranging terms)}$$

$$\geq (1-\varepsilon)\left(\frac{1}{k}\left(1-\frac{1}{k}\right)^{k-(t+1)}f(\mathsf{OPT}) - \frac{1}{k}c(\mathsf{OPT})\right) \quad \text{(Non-negativity of $f$)}$$

First, assume that $\beta$ is a multiple of $1/k$. We lower bound the first $k - \beta \cdot k$ terms of $\sum_{t=0}^{k-1}\mathbb{E}[\Phi_{t+1}(S_{t+1}) - \Phi_t(S_t)]$ by 0 and the last $\beta \cdot k$ terms by the previous inequality. Thus, we get

$$\sum_{t=0}^{k-1}\mathbb{E}[\Phi_{t+1}(S_{t+1}) - \Phi_t(S_t)] \geq \sum_{t=k-\beta\cdot k}^{k-1}\mathbb{E}[\Phi_{t+1}(S_{t+1}) - \Phi_t(S_t)]$$

$$\geq \sum_{t=k-\beta\cdot k}^{k-1}(1-\varepsilon)\left\{\frac{1}{k}\left(1-\frac{1}{k}\right)^{k-(t+1)}f(\mathsf{OPT}) - \frac{1}{k}c(\mathsf{OPT})\right\}$$

$$\geq (1-\varepsilon)\left(\sum_{t=k-\beta\cdot k}^{k-1}\frac{1}{k}\left(1-\frac{1}{k}\right)^{k-(t+1)}\right)f(\mathsf{OPT}) - \beta \cdot c(\mathsf{OPT}) \quad \text{(Non-negativity of $c$)}$$

---

[3]In fact, it holds for the realization of the random variables and not just in expectation.

$$= \frac{1-\varepsilon}{k} \cdot \frac{1 - \left(1 - \frac{1}{k}\right)^{\beta \cdot k}}{1 - \left(1 - \frac{1}{k}\right)} \cdot f(\text{OPT}) - \beta \cdot c(\text{OPT}) \qquad \text{(Sum of geometric series)}$$

$$\geq (1-\varepsilon)(1 - e^{-\beta}) \cdot f(\text{OPT}) - \beta \cdot c(\text{OPT}). \qquad \text{(Proposition B.4)}$$

In case $\beta$ is not a multiple of $1/k$, the same analysis goes through with $\hat{\beta}$ being the smallest multiple of $1/k$ that is greater than $\beta$ and the guarantee we get is $((1-\varepsilon)(1 - e^{-\hat{\beta}}), \hat{\beta})$, which is at least $((1-\varepsilon)(1 - e^{-\beta}), \beta + 1/k)$. $\qquad \square$

## B.3. Noisy Setting

Following the model of Horel & Singer (2016), in this section we discuss adaptations of our results to the noisy setting where we have access to an oracle $F : 2^{\mathcal{N}} \to \mathbb{R}_{\geq 0}$ so that

$$(1-\varepsilon)f(S) \leq F(S) \leq (1+\varepsilon)f(S), \forall S \subseteq 2^{\mathcal{N}},$$

for some $\varepsilon > 0$. Our Algorithm 7 is an adaptation of Harshaw et al. (2019) and Gong et al. (2023) with the main difference being that when we evaluate the score of an element we look at its minimum marginal contribution across all the sets we have constructed in the history of the execution. This is because the function $F$ is not submodular.

Similarly as in the noiseless setting, we let

$$\tilde{\Phi}_t(T) = \left(1 - \frac{1}{k}\right)^{k-t} F(T) - x \cdot \sum_{j \in T} c_j, \qquad\qquad \forall T \subseteq \mathcal{N},$$

$$\tilde{\Psi}_t(\mathcal{T}, i) = \max\left\{0, \left(1 - \frac{1}{k}\right)^{k-(t+1)} \min_{S \in \mathcal{T}} F(i|S) - x \cdot c_i\right\}, \qquad\qquad \forall \mathcal{T} \subseteq 2^{\mathcal{N}}, i \in \mathcal{N}.$$

---

**Algorithm 7** Noisy distorted greedy algorithm

---

**Data:** A set of $n$ items $\mathcal{N}$, a monotone submodular function $f : 2^{\mathcal{N}} \to \mathbb{R}_{\geq 0}$, a cost function $c : \mathcal{N} \to \mathbb{R}_{\geq 0}$, a capacity constraint $k \leq n$, a cost parameter $x$
**Result:** A subset of the items $R \subseteq \mathcal{N}$ with $|S| \leq k$
$S_0 = \emptyset$ $\mathcal{S}_0 = S_0$ **for** $j$ *from* $1$ *to* $k$ **do**
$\quad\big|\quad$ $G(i, S_{j-1}, \boldsymbol{c}, j, r) = \left(1 - \frac{1}{n}\right)^{k-j} \cdot \min_{1 \leq t \leq j} F(i \mid S_{t-1}) - x \cdot c_i, \forall i \in S$ $i^* = \arg\max_{i \notin S_{j-1}} G(i, S_{j-1}, \boldsymbol{c}, j, r)$ **if**
$\quad\big|\quad$ $G(i_t, S_{j-1}, \boldsymbol{c}, j, r) > 0$ **then**
$\quad\big|\quad\big|\quad$ $S_j = S_{j-1} \cup \{i^*\}$ $\mathcal{S}_j = \mathcal{S}_{j-1} \cup S_j$
$\quad\big|$ **end**
$\quad\big|$ **else**
$\quad\big|\quad\big|\quad$ $S_j = S_{j-1}$ $\mathcal{S}_j = \mathcal{S}_{j-1}$
$\quad\big|$ **end**
**end**
**return** $S_n$

---

The next result describes the connection between $\tilde{\Phi}, \tilde{\Psi}$. Our proof is an adaptation of Harshaw et al. (2019); Gong et al. (2023).

**Lemma B.7.** *In each iteration $t \in \{0, \ldots, k-1\}$ of Algorithm 7 it holds that*

$$\tilde{\Phi}_{t+1}(S_{t+1}) - \tilde{\Phi}_t(S_t) \geq \tilde{\Psi}_t(\mathcal{S}_t, i_t) + \frac{1}{k}\left(1 - \frac{1}{k}\right)^{k-(t+1)} F(S_t),$$

*where $\mathcal{S}_t = \{S_0, \ldots, S_t\}$.*

*Proof.* By definition we have that

$$\tilde{\Phi}_{t+1}(S_{t+1}) - \tilde{\Phi}_t(S_t) = \left(1 - \frac{1}{k}\right)^{k-(t+1)} F(S_{t+1}) - c(S_{t+1}) - \left(1 - \frac{1}{k}\right)^{k-t} F(S_t) - c(S_t)$$

$$= \left(1 - \frac{1}{k}\right)^{k-(t+1)} F(S_{t+1}) - c(S_{t+1}) - \left(1 - \frac{1}{k}\right)^{k-(t+1)} \left(1 - \frac{1}{k}\right) F(S_t) - c(S_t)$$

$$= \left(1 - \frac{1}{k}\right)^{k-(t+1)} (F(S_{t+1}) - F(S_t)) - (c(S_{t+1}) - c(S_t)) + \frac{1}{k}\left(1 - \frac{1}{k}\right)^{k-(t+1)} F(S_t).$$

Now we consider two cases. If $S_{t+1} = S_t$ then $\tilde{\Psi}(S_t, i_t) = 0$ and the inequality holds. Otherwise, we have

$$\left(1 - \frac{1}{k}\right)^{k-(t+1)} (F(S_{t+1}) - F(S_t)) - (c(S_{t+1}) - c(S_t)) + \frac{1}{k}\left(1 - \frac{1}{k}\right)^{k-(t+1)} F(S_t) =$$

$$\left(1 - \frac{1}{k}\right)^{k-(t+1)} F(i_t|S_t) - c_{i_t} + \frac{1}{k}\left(1 - \frac{1}{k}\right)^{k-(t+1)} F(S_t) \geq$$

$$\left(1 - \frac{1}{k}\right)^{k-(t+1)} \min_{1 \leq j \leq t} F(i_t|S_j) - c_{i_t} + \frac{1}{k}\left(1 - \frac{1}{k}\right)^{k-(t+1)} F(S_t) =$$

$$\tilde{\Psi}_t(S_t, i_t) + \frac{1}{k}\left(1 - \frac{1}{k}\right)^{k-(t+1)} F(S_t).$$

$\square$

The next result relates the marginal gain in each iteration of Algorithm 7 to the function $\tilde{\Psi}_t(S_t, i_t)$. It builds upon the approach of Harshaw et al. (2019); Gong et al. (2023).

**Lemma B.8.** *In each iteration of Algorithm 7 it holds that*

$$\tilde{\Psi}_t(S_t, i_t) \geq \frac{1-\varepsilon}{k}\left(1 - \frac{1}{k}\right)^{k-(t+1)} (f(\mathsf{OPT}) - f(S_t)) - 2\varepsilon\left(1 - \frac{1}{k}\right)^{k-(t+1)} f(S_k) - \frac{x}{k}c(\mathsf{OPT}).$$

*Proof.* First, notice that

$$F(S|i) = F(S \cup \{i\}) - F(S) \geq (1-\varepsilon)f(S \cup \{i\}) - (1+\varepsilon)f(S) = (1-\varepsilon)f(S|i) - 2\varepsilon f(S). \tag{1}$$

Let OPT be the optimal solution of $\max_S\{f(S) - c(S)\}$. We have that

$$k\tilde{\Psi}_t(S_t, i_t) = k \cdot \max\left\{0, \left(1 - \frac{1}{k}\right)^{k-(t+1)} \min_{S \in \mathcal{S}_t} F(i_t|S) - x \cdot c_i\right\} \qquad \text{(Definition)}$$

$$\geq k \cdot \left(\left(1 - \frac{1}{k}\right)^{k-(t+1)} \min_{S \in \mathcal{S}_t} F(i_t|S) - x \cdot c_i\right) \qquad \text{(Restricting max)}$$

$$= k \cdot \max_{i \in \mathcal{N}}\left\{\left(1 - \frac{1}{k}\right)^{k-(t+1)} \min_{S \in \mathcal{S}_t} F(i|S) - x \cdot c_i\right\} \qquad \text{(Definition)}$$

$$\geq |\mathsf{OPT}| \cdot \max_{i \in \mathcal{N}}\left\{\left(1 - \frac{1}{k}\right)^{k-(t+1)} \min_{S \in \mathcal{S}_t} F(i|S) - x \cdot c_i\right\} \qquad (k \geq |\mathsf{OPT}|)$$

$$\geq |\mathsf{OPT}| \cdot \max_{i \in \mathsf{OPT}}\left\{\left(1 - \frac{1}{k}\right)^{k-(t+1)} \min_{S \in \mathcal{S}_t} F(i|S) - x \cdot c_i\right\} \qquad \text{(Restricting max)}$$

$$\geq \sum_{i \in \mathsf{OPT}}\left\{\left(1 - \frac{1}{k}\right)^{k-(t+1)} \min_{S \in \mathcal{S}_t} F(i|S) - x \cdot c_i\right\} \qquad \text{(Averaging argument)}$$

$$= \sum_{i \in \mathsf{OPT}}\left\{\left(1 - \frac{1}{k}\right)^{k-(t+1)} F(i|S^i) - x \cdot c_i\right\} \qquad (S^i = \arg\min_{S \in \mathcal{S}_t} F(i|S))$$

$$\geq \sum_{i \in \mathsf{OPT}}\left\{\left(1 - \frac{1}{k}\right)^{k-(t+1)} ((1-\varepsilon)f(i|S^i) - 2\varepsilon f(S^i)) - x \cdot c_i\right\} \qquad \text{(Equation (1))}$$

$$\geq \sum_{i \in \text{OPT}} \left\{ \left(1 - \frac{1}{k}\right)^{k-(t+1)} \left((1-\varepsilon)f(i|S_t) - 2\varepsilon f(S^i)\right) - x \cdot c_i \right\} \qquad \text{(Submodularity of } f\text{)}$$

$$\geq \sum_{i \in \text{OPT}} \left\{ \left(1 - \frac{1}{k}\right)^{k-(t+1)} (1-\varepsilon)f(i|S_t) - 2\varepsilon \left(1 - \frac{1}{k}\right)^{k-(t+1)} f(S_k) - x \cdot c_i \right\}. \qquad \text{(Monotonicity of } f\text{)}$$

We will bound each term of the summation separately. Notice that $\sum_{i \in \text{OPT}} x \cdot c_i = x \cdot c(\text{OPT})$. Similarly, $\sum_{i \in \text{OPT}} 2\varepsilon \left(1 - \frac{1}{k}\right)^{k-(t+1)} f(S_k) \leq 2\varepsilon |\text{OPT}| f(S_k) \leq 2\varepsilon k f(S_k)$, and by the submodularity and monotonicity of $f$ we have

$$\sum_{i \in \text{OPT}} (1-\varepsilon)f(i|S_t) \geq (1-\varepsilon)\left(f(S_t \cup \text{OPT}) - f(S_t)\right) \geq (1-\varepsilon)\left(f(\text{OPT}) - f(S_t)\right).$$

Putting everything together, we get

$$k\tilde{\Psi}_t(\mathcal{S}_t, i_t) \geq (1-\varepsilon)\left(1 - \frac{1}{k}\right)^{k-(t+1)} (f(\text{OPT}) - f(S_t)) - 2\varepsilon k \left(1 - \frac{1}{k}\right)^{k-(t+1)} f(S_k) - xc(\text{OPT}),$$

and dividing by $k$ we get

$$\tilde{\Psi}_t(\mathcal{S}_t, i_t) \geq \frac{1-\varepsilon}{k}\left(1 - \frac{1}{k}\right)^{k-(t+1)} (f(\text{OPT}) - f(S_t)) - 2\varepsilon \left(1 - \frac{1}{k}\right)^{k-(t+1)} f(S_k) - \frac{x}{k}c(\text{OPT}).$$

$\square$

We are now ready to prove the bi-criteria guarantees of Algorithm 7. Our analysis follows Harshaw et al. (2019); Gong et al. (2023).

**Theorem B.9.** *For* $x = 1 + 2\varepsilon k + \varepsilon$ *Algorithm 7 returns a set* $S_k$ *with*

$$f(S_k) - c(S_k) \geq \frac{1-\varepsilon}{1 + 2\varepsilon k + \varepsilon} \cdot (1 - 1/e) f(\text{OPT}) - c(S_k).$$

*Proof.* Combining Lemma B.7 and Lemma B.8 we immediately get that

$$\tilde{\Phi}_{t+1}(S_{t+1}) - \tilde{\Phi}_t(S_t) \geq$$

$$\tilde{\Psi}_t(\mathcal{S}_t, i_t) + \frac{1}{k}\left(1 - \frac{1}{k}\right)^{k-(t+1)} F(S_t) \geq$$

$$\frac{1-\varepsilon}{k}\left(1 - \frac{1}{k}\right)^{k-(t+1)} (f(\text{OPT}) - f(S_t)) - 2\varepsilon \left(1 - \frac{1}{k}\right)^{k-(t+1)} f(S_k) - \frac{x}{k}c(\text{OPT}) + \frac{1}{k}\left(1 - \frac{1}{k}\right)^{k-(t+1)} F(S_t) \geq$$

$$\frac{1-\varepsilon}{k}\left(1 - \frac{1}{k}\right)^{k-(t+1)} (f(\text{OPT}) - f(S_t)) - 2\varepsilon \left(1 - \frac{1}{k}\right)^{k-(t+1)} f(S_k) - \frac{x}{k}c(\text{OPT}) + \frac{1-\varepsilon}{k}\left(1 - \frac{1}{k}\right)^{k-(t+1)} f(S_t) =$$

$$\frac{1-\varepsilon}{k}\left(1 - \frac{1}{k}\right)^{k-(t+1)} f(\text{OPT}) - 2\varepsilon \left(1 - \frac{1}{k}\right)^{k-(t+1)} f(S_k) - \frac{x}{k}c(OPT).$$

Moreover, notice that a straightforward bound is

$$\tilde{\Phi}_{t+1}(S_{t+1}) - \tilde{\Phi}_t(S_t) \geq 0.$$

By definition of $\tilde{\Phi}$ we have that

$$\tilde{\Phi}_0(S_0) = \left(1 - \frac{1}{k}\right)^k F(\emptyset) - xc(\emptyset) = 0$$

$$\tilde{\Phi}_k(S_k) = \left(1 - \frac{1}{k}\right)^k F(S_k) - xc(S_k).$$

Combining the previous inequalities we get

$$
\begin{aligned}
F(S_k) &- xc(S_k) \\
&= \tilde{\Phi}_k(S_k) - \tilde{\Phi}_0(S_0) \\
&= \sum_{i=1}^{k} (\Phi_i(S_i) - \tilde{\Phi}_{i-1}(S_{i-1})) \\
&\geq \sum_{i=1}^{k} \left( \frac{1-\varepsilon}{k} \left(1 - \frac{1}{k}\right)^{k-(t+1)} f(\mathsf{OPT}) - 2\varepsilon \left(1 - \frac{1}{k}\right)^{k-(t+1)} f(S_k) - \frac{x}{k} c(OPT) \right) \\
&= \frac{1-\varepsilon}{k} \sum_{i=1}^{k} \left(1 - \frac{1}{k}\right)^{k-(t+1)} f(\mathsf{OPT}) - 2\varepsilon \sum_{i=1}^{k} \left(1 - \frac{1}{k}\right)^{k-(t+1)} f(S_k) - x \cdot c(\mathsf{OPT}) \\
&= \frac{1-\varepsilon}{k} \cdot k \cdot \left(1 - (1-1/k)^k\right) f(\mathsf{OPT}) - 2\varepsilon k \cdot \left(1 - (1-1/k)^k\right) f(S_k) - x \cdot c(\mathsf{OPT}),
\end{aligned}
$$

which implies that

$$
\begin{aligned}
(1+\varepsilon)f(S_k) - xc(S_k) &\geq (1-\varepsilon) \cdot \left(1 - (1-1/k)^k\right) f(\mathsf{OPT}) - 2\varepsilon k \cdot \left(1 - (1-1/k)^k\right) f(S_k) - xc(\mathsf{OPT}) \\
&\geq (1-\varepsilon) \cdot \left(1 - (1-1/k)^k\right) f(\mathsf{OPT}) - 2\varepsilon k \cdot f(S_k) - xc(\mathsf{OPT}),
\end{aligned}
$$

and rearranging we get

$$
\begin{aligned}
(1 + 2\varepsilon k + \varepsilon)f(S_k) - xc(S_k) &\geq (1-\varepsilon) \cdot \left(1 - (1-1/k)^k\right) f(\mathsf{OPT}) - xc(\mathsf{OPT}) \\
&\geq (1-\varepsilon) \cdot (1 - 1/e) f(\mathsf{OPT}) - xc(\mathsf{OPT}).
\end{aligned}
$$

Finally, we can simplify $x = 1 + 2\varepsilon k + \varepsilon$ and get

$$
f(S_k) - c(S_k) \geq \frac{1-\varepsilon}{1 + 2\varepsilon k + \varepsilon} \cdot (1 - 1/e) f(\mathsf{OPT}) - c(S_k).
$$

$\square$

*Remark* B.10 (IC, IR, NAS in the Noisy Setting). We remark that our modification in Algorithm 7 enforces the diminishing returns property in the scores of the elements that are added to the constructed solutions. Hence, an identical argument to the one we used in the proof of Theorem 4.3 shows that the NAS property is satisfied. The IC, IR properties continue to hold since they are not affected by the submodularity of the function.

## C. Omitted Details from Section 4.1

Let $\mathsf{OPT}(\boldsymbol{b}) \in \arg\max_{S \in 2^{\mathcal{N}}} f(S) - \sum_{i \in S} c_i$, i.e., the optimal solution to the optimization problem when sellers report bids $b$. Given a bid profile $\boldsymbol{b}$, the VCG mechanism purchases items from the sellers in $\mathsf{OPT}(\boldsymbol{b})$ and the payment $p_i$ to each $i \in \mathsf{OPT}(\boldsymbol{b})$ is given by

$$
p_i = \left( f\big(\mathsf{OPT}(\boldsymbol{b})\big) - \sum_{j \in \mathsf{OPT}(\boldsymbol{b}) \setminus \{i\}} c_j \right) - \left( f\big(\mathsf{OPT}((\infty, \boldsymbol{b}_{-i}))\big) - \sum_{j \in \mathsf{OPT}((\infty, \boldsymbol{b}_{-i}))} c_j \right).
$$

*Proof of Proposition 4.1.* We would like to show $f\big(\mathsf{OPT}(\boldsymbol{b})\big) \geq \sum_{i \in \mathsf{OPT}(\boldsymbol{b})} p_i$ when $f$ is a submodular function.

$$
\begin{aligned}
\sum_{i \in \mathsf{OPT}(\boldsymbol{b})} p_i &= \sum_{i \in \mathsf{OPT}(\boldsymbol{b})} \left( f\big(\mathsf{OPT}(\boldsymbol{b})\big) - \sum_{j \in \mathsf{OPT}(\boldsymbol{b}) \setminus \{i\}} c_j \right) - \left( f\big(\mathsf{OPT}((\infty, \boldsymbol{b}_{-i}))\big) - \sum_{j \in \mathsf{OPT}((\infty, \boldsymbol{b}_{-i}))} c_j \right) \\
&\leq \sum_{i \in \mathsf{OPT}(\boldsymbol{b})} \left( f\big(\mathsf{OPT}(\boldsymbol{b})\big) - \sum_{j \in \mathsf{OPT}(\boldsymbol{b}) \setminus \{i\}} c_j \right) - \left( f\big(\mathsf{OPT}(\boldsymbol{b}) \setminus \{i\}\big) - \sum_{j \in \mathsf{OPT}(\boldsymbol{b}) \setminus \{i\}} c_j \right)
\end{aligned}
$$

$$= \sum_{i \in \mathsf{OPT}(\boldsymbol{b})} f(\mathsf{OPT}(\boldsymbol{b})) - f(\mathsf{OPT}(\boldsymbol{b}) \setminus \{i\})$$

$$\leq f(\mathsf{OPT}(\boldsymbol{b})),$$

where the first inequality uses the fact that $\mathsf{OPT}((\infty, \boldsymbol{b}_{-i})) \in \arg\max_{S:i \notin S} f(S) - \sum_{j \in S} c_j$ and the second inequality follows the property of submodularity. $\qquad \square$

*Proof of Theorem 4.3.* Fix $\boldsymbol{b}_{-i}$ and $r$, and let $\left\{ S_0^{b_i}, S_1^{b_i}, \cdots, S_n^{b_i} \right\}$ be the intermediate tentative solutions of running $\mathcal{A}(\mathcal{N}, (b_i, \boldsymbol{b}_{-i}), r)$. Moreover, let $p_{i,k} = \max_{1 \leq j \leq k} z_j^*$ where

$$z_j^* = \sup \left\{ z \mid i = \arg\max_{\ell \notin S_{j-1}^\infty} G\left(\ell, S_{j-1}^\infty, (z, \boldsymbol{b}_{-i}), j, r\right) \ \& \ G\left(i, S_{j-1}^\infty, (z, \boldsymbol{b}_{-i}), j, r\right) > 0 \right\}.$$

We will show that for any $i$, $\boldsymbol{b}_{-i}$, $r$, and $k$: (1) if $b_i < p_{i,k}$, $i \in S_k^{b_i}$, and if $b_i > p_{i,k}$, $i \notin S_k^{b_i}$; (2) if $i \notin S_k^{b_i}$, $S_k^{b_i} = S_k^\infty$. We prove it by induction on $k$. When $k = 1$, from Algorithm 1, the definition of $p_{i,1}$, and Assumption 4.2(1), we have

- when $b_i < p_{i,1}$, then $S_1^{b_i} = \{i\}$ since $i = \arg\max_\ell G(\ell, \emptyset, (b_i, \boldsymbol{b}_{-i}), 1, r)$;

- when $b_i > p_{i,1}$, with $\ell_1^* = \arg\max_{\ell \neq i} G(\ell, \emptyset, (b_i, \boldsymbol{b}_{-i}), 1, r)$, if $G(\ell_1^*, \emptyset, (b_i, \boldsymbol{b}_{-i}), 1, r) > 0$, $S_1^{b_i} = \{\ell_1^*\}$; otherwise, $S_1^{b_i} = \emptyset$.

Moreover, notice that whenever $i \neq \ell_1^*$, $i$'s bid becomes irrelevant due to Assumption 4.2(3) so that

$$\ell_1^* = \arg\max_\ell G(\ell, \emptyset, (\infty, \boldsymbol{b}_{-i}), 1, r),$$

and therefore, if $i \notin S_1^{b_i}$, $S_1^{b_i} = S_1^\infty$. For the inductive step, we assume the previous arguments hold for all rounds up to $k$. Then, for round $k + 1$, we have

- when $b_i < p_{i,k+1}$, then either we have $i \in S_k^{b_i} \subset S_{k+1}^{b_i}$ or we have $S_k^{b_i} = S_k^\infty$, which also implies $i \in S_{k+1}^{b_i}$ since $i = \arg\max_{\ell \notin S_k^\infty} G(\ell, S_k^\infty, (b_i, \boldsymbol{b}_{-i}), k + 1, r)$ and $G(i, S_k^\infty, (b_i, \boldsymbol{b}_{-i}), k + 1, r) > 0$;

- when $b_i > p_{i,k+1}$, with $\ell_{k+1}^* = \arg\max_{\ell \neq i} G(\ell, S_k^\infty, (b_i, \boldsymbol{b}_{-i}), k + 1, r)$, if $G(\ell_{k+1}^*, S_k^\infty, (b_i, \boldsymbol{b}_{-i}), k + 1, r) > 0$, $S_{k+1}^{b_i} = S_k^\infty \cup \{\ell_{k+1}^*\}$; otherwise, $S_{k+1}^{b_i} = S_k^\infty$.

Again, notice that whenever $i \neq \ell_{k+1}^*$, $i$'s bid becomes irrelevant due to Assumption 4.2(3) so that

$$\ell_{k+1}^* = \arg\max_{\ell \notin S_k^\infty} G(i, S_k^\infty, (\infty, \boldsymbol{b}_{-i}), k + 1, r);$$

and therefore, if $i \notin S_{k+1}^{b_i}$, $S_{k+1}^{b_i} = S_{k+1}^\infty$, which concludes the inductive step. Observe that Algorithm 2 exactly computes the critical bid $p_i = p_{i,n}$ for seller $i$ such that if $b_i < p_{i,n}$, $i \in S^*$ and if $b_i > p_{i,n}$, $i \notin S^*$. From Myerson's lemma (Myerson, 1981), the mechanism is IC for seller $i$. Moreover, the mechanism is IR because $i \in S^*$ only if $c_i = b_i \leq p_{i,n}$ when seller $i$ reports truthfully.

Finally, to prove the mechanism satisfies NAS, assume seller $i$ is added to the solution in round $k$ such that $S_k^{b_i} \setminus S_{k-1}^{b_i} = \{i\}$. From Assumption 4.2(2), we have $b_i \leq f(S_k^{b_i}) - f(S_{k-1}^{b_i})$ in order to have a positive score for $i$ in round $k$. As we have established that if $b_i < p_{i,k-1}$, then $i \in S_{k-1}^{b_i}$, we have $p_{i,k-1} \leq b_i \leq f(S_k^{b_i}) - f(S_{k-1}^{b_i})$. For $j \geq k$, we have $z_j^* \leq f(i \mid S_{j-1}^\infty)$ due to the fact that $G(i, S_{j-1}, \boldsymbol{b}, j, r) < 0$ whenever $b_i > f(i \mid S_{j-1}^\infty)$ from Assumption 4.2(2). From the previously proved fact that if $i \notin S_{k-1}^{b_i}$, $S_{k-1}^{b_i} = S_{k-1}^\infty$, we have $S_{k-1}^{b_i} \subseteq S_{j-1}^\infty$ for $j \geq k$ by submodularity of $f$. As a result, for $j \geq k$, we have that

$$z_j^* \leq f(i \mid S_{j-1}^\infty) \leq f(i \mid S_{k-1}^{b_i}) = f(S_k^{b_i}) - f(S_{k-1}^{b_i}),$$

which implies that $p_i = \max\left\{p_{i,k-1}, \max_{k \le j \le n} z_j^*\right\} \le f(S_k^{b_i}) - f(S_{k-1}^{b_i})$. Thus,

$$\sum_{i \in S_n^{b_i}} p_i \le \sum_{k=1}^{n} f(S_k^{b_i}) - f(S_{k-1}^{b_i}) = f(S_n^{b_i}).$$

$\square$

### C.1. Omitted Details from Online Mechanism Design Framework

---

**Algorithm 8** A posted-price mechanism construction for a given meta algorithm $\mathcal{A}^o$

---

**Data:** A set of sellers arriving online and a meta algorithm $\mathcal{A}^o$
**Result:** A subset of sellers to purchase from and a vector of payment to sellers
Generate a random seed $r$ if needed or set $r = 0$
$S_0 = \emptyset, k = 0$
**while** *there exists a newly arrived seller $k + 1$* **do**
  $k = k + 1$
  Let $\hat{p}_k$ be the unique solution of the equation $G\left(k, S_{k-1}, \left(\mathbf{c}_{(1,k-1)}, z\right), r\right) = 0$ in terms of $z$
  Post price $\hat{p}_k$ to seller $k$
  **if** *Seller $k$ accepts the posted price* **then**
    $S_k = S_{k-1} \cup \{k\}$
    $p_k = \hat{p}_k$
  **end**
  **else**
    $S_k = S_{k-1}$
    $p_k = 0$
  **end**
**end**
**return** $S_k$ *and* $\mathbf{p}$

---

*Proof of Theorem 4.6.* The IC and IR properties follow immediately from the fact that the mechanism is a posted-price mechanism, such that seller $k$ accepts posted price $\hat{p}_k$ if and only if $\hat{p}_k > c_k$[4]. From the definition of $\hat{p}_k$ and the monotonicity of $G$ from Assumption 4.5(1), we have $G(k, S_{k-1}, \mathbf{c}_{(1,k)}, r) > 0$ if and only if $c_k < \hat{p}_k$, and therefore, Algorithm 8 and Algorithm 3 return the same solution if sellers always best respond to the posted prices.

Finally, we prove for NAS. For seller $k \in S^*$, from Assumption 4.5(2), we have

$$\hat{p}_k \le f(k \mid S_{k-1}) = f(S_k) - f(S_{k-1}).$$

Thus, summing up over all sellers in $S^*$, we have

$$\sum_{k \in S^*} p_k \le \sum_{k \in S^*} f(S_k) - f(S_{k-1}) = f(S^*).$$

$\square$

## D. Omitted Details from Section 5

*Proof of Theorem 5.1.* We construct an instance with $L + 2$ sellers, indexed by $\{1, \cdots, L, L+1, L+2\}$, with a submodular function $f$ such that: if $\{L + 1, L + 2\} \cap S = \emptyset$, $f(S) = |S|$; otherwise, $f(S) = L$. Moreover, let $b_i = 1/L$ for $i \le L$ and $b_{L+1} = b_{L+2} = L - 2$. We next describe the strategy of the adversary for selecting seller $i \in S \setminus \mathcal{D}(S, \mathbf{p})$ in line 4 of Algorithm 4.

First, the adversary keeps selecting a seller $j \in \{L + 1, L + 2\}$ until either $p_{L+1} < L - 1$ or $p_{L+2} < L - 1$. This is achievable since if $\{L + 1, L + 2\} \subseteq \mathcal{D}(S, \mathbf{p})$, the welfare is at most $L - 2 \times (L - 1) < 0$, so that the approximation

---

[4]For simplicity, we assume that the seller does not accept the offer when the cost is exactly the same as the posted price.

guarantee is violated. Without loss of generality, assume that $p_{L+1} < L - 1$, and therefore, the welfare of selecting seller $L + 1$ alone is $f(\{L + 1\}) - p_{L+1} > 1$.

Next, the adversary iterates over sellers from seller 1 to seller $L$ such that for each seller $i$, keep selecting the seller $i$ until $i$ leaves the market, i.e., $p_i < b_i$. We argue that the above process is achievable as the welfare of selecting any subset containing seller $i$ is at most 1.

The final welfare is at most 2 by purchasing items from either seller $L + 1$ or $L + 2$. However, the optimal welfare is obtained by purchasing from sellers in $\{1, \cdots, L\}$, which gives welfare $L - 1$. $\qquad \square$

*Proof of Theorem 5.2.* The $(\frac{1}{2}, 1)$-approximate demand oracle $\hat{\mathcal{D}}$ we construct maintains a tentative solution $S$ initialized at $S = \emptyset$. For each iteration, let $i$ be the candidate selected by the adversary in the previous iteration. Update $S = S \cup \{i\}$ if $f(i \mid S) > 2p_i$ (otherwise, keep $S$ as it is), and return $S$. Let $S^*$ be the subset returned by the Algorithm 4 with demand oracle $\hat{\mathcal{D}}$. For each seller $i \in S^*$, let $\hat{p}_i$ and $S_i$ be the price $p_i$ and the tentative solution $S$ maintained by the demand oracle right before $i$ is added to the solution, respectively. For seller $i \notin S^*$, let $\hat{p}_i$ and $S_i$ be the price $p_i$ and the tentative solution $S$ maintained by the demand oracle when $i$ is removed from the descending auction, respectively.

Let $g(S) = f(S) - 2 \cdot \sum_{i \in S} b_i$, which is also a submodular function. For any seller $i \in \mathsf{OPT} \setminus S^*$,

$$g(i \mid S^*) = f(i \mid S^*) - 2 \cdot b_i \leq f(i \mid S^*) - 2 \cdot \hat{p}_i \leq 2\varepsilon$$

where the first inequality follows $b_i - \varepsilon \leq \hat{p}_i < b_i$ for $i \notin S^*$ and the second inequality follows the fact that $f(i \mid S^*) \leq f(i \mid S_i) \leq 2 \cdot (\hat{p}_i + \varepsilon)$. As a result, we have

$$2\varepsilon \cdot |\mathsf{OPT} \setminus S^*| \geq \sum_{i \in \mathsf{OPT} \setminus S^*} g(i \mid S^*) \geq g(S^* \cup \mathsf{OPT}) - g(S^*)$$

$$= \left( f(S^* \cup \mathsf{OPT}) - f(S^*) \right) - 2 \cdot \sum_{i \in \mathsf{OPT} \setminus S^*} b_k$$

$$\geq f(\mathsf{OPT}) - f(S^*) - 2 \cdot \sum_{i \in \mathsf{OPT}} b_i,$$

where the second inequality follows submodularity of $g$. Rearranging, we obtain

$$f(S^*) \geq f(\mathsf{OPT}) - 2 \cdot \sum_{i \in \mathsf{OPT}} b_i - 2\varepsilon \cdot |\mathsf{OPT} \setminus S^*| \geq f(\mathsf{OPT}) - 2 \cdot \sum_{i \in \mathsf{OPT}} b_i - 2n\varepsilon.$$

Observe that we have

$$f(S^*) - 2 \cdot \sum_{i \in S^*} b_i = \sum_{i \in S^*} f(i \mid S_i) - 2 \cdot b_i \geq \sum_{i \in S^*} f(i \mid S_i) - 2 \cdot \hat{p}_i \geq 0,$$

where the first inequality follows $\hat{p}_i \geq b_i$ for $i \in S^*$ and the second inequality is due to $i \in S^*$ and the definition of $\hat{p}_i$ and $S_i$. Therefore, we have $\sum_{i \in S^*} b_i \leq \frac{1}{2} \cdot f(S^*)$, indicating $f(S^*) - \sum_{i \in S^*} b_i \geq \frac{1}{2} \cdot f(S^*)$. Putting everything together, we have

$$f(S^*) - \sum_{i \in S^*} b_i \geq \frac{1}{2} \cdot f(S^*) \geq \frac{1}{2} \cdot f(\mathsf{OPT}) - \sum_{i \in \mathsf{OPT}} b_i - n\varepsilon.$$

$\qquad \square$

# E. Omitted Details from Section 6

**Heuristic Implementation**   We provide a heuristic to optimize the running time of Algorithm 1 if the scoring rule of a meta algorithm $\mathcal{A} = (G)$ has a diminishing-return structure, i.e., $G(i, S, \boldsymbol{b}, j, r) \geq G(i, T, \boldsymbol{b}, k, r)$ for all $S \subseteq T$ and $j \leq k$. In particular, inspired by the lazy implementation of the classical greedy algorithm (Minoux, 2005), we maintain a priority queue that records each candidate's *last-updated score*. The priority queue is initialized consisting of elements with key $i$ and value $G(i, \emptyset, \boldsymbol{c}, 0, r)$ for each $i \in \mathcal{N}$. For each iteration $k$, we repeatedly compare candidate $i_1$ with the highest score in $Q$ and candidate $i_2$ with the second-highest score in $Q$. If $G(i_1, S, \boldsymbol{c}, k, r)$ for the tentative solution $S$ is positive and is

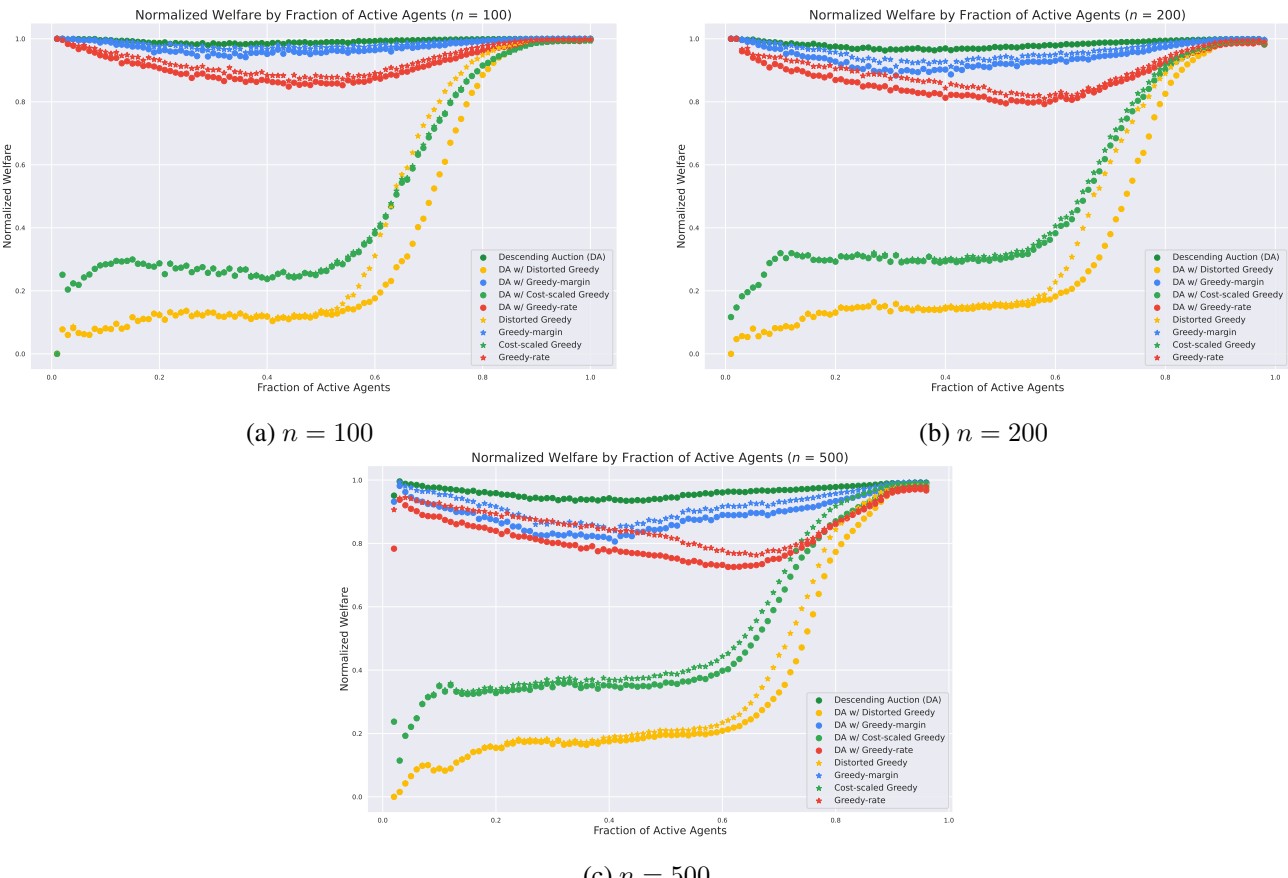

(a) $n = 100$

(b) $n = 200$

(c) $n = 500$

*Figure 3.* Welfare as a function of the fraction of active agents for $n \in \{100, 200, 500\}$.

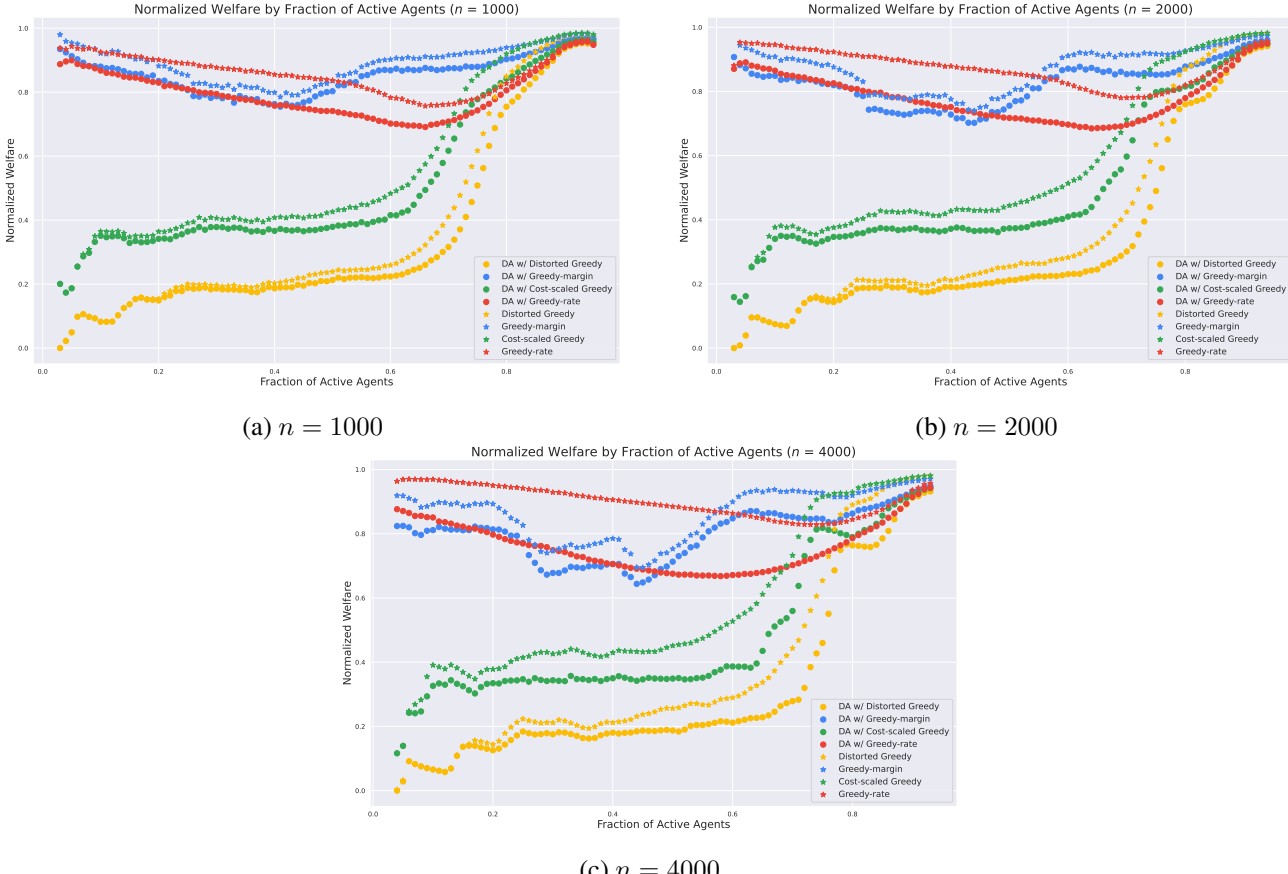

(a) $n = 1000$

(b) $n = 2000$

(c) $n = 4000$

*Figure 4.* Welfare as a function of the fraction of active agents for $n \in \{1000, 2000, 4000\}$.

---

**Algorithm 9** Descending auction construction for a given meta algorithm $\mathcal{A}^o$ and a step size of $\varepsilon$

---

**Data:** A set of seller $\mathcal{N}$ and a bid profile $\boldsymbol{b}$ from sellers
**Result:** A subset $S^*$ of sellers to purchase from and a vector $\boldsymbol{p}$ of payment to sellers
Generate a random seed $r$ if needed or set $r = 0$   $S_0 = \emptyset$   Set initial prices as $p_i = f(i \mid \emptyset)$   **for** $k$ *from* 1 *to* $n$ **do**

> Let $\hat{p}_k$ be the unique solution of the equation $G\left(k, S_{k-1}, \left(\boldsymbol{c}_{(1,k-1)}, z\right), r\right) = 0$ in terms of $z$   Update $p_k = \hat{p}_k$   **if**
>
> > $p_k > b_i$ **then**
> > | $S_k = S_{k-1} \setminus \{i\}$
> > **end**
> > **else**
> > | $S_k = S_{k-1}$  $p_k = 0$
> > **end**

**end**
**return** $S$ *and* $\boldsymbol{p}$

---

larger than the $i_2$'s score maintained in $Q$, then $i_1$ is guaranteed to have the highest score due to the diminishing-return structure; otherwise, update the score for $i_1$ and repeat (see Algorithm 10 in Appendix). The running time of the payment calculation in Algorithm 2 can be optimized in a similar way (see Algorithm 11 in Appendix for details). Note that the scoring rules of all deterministic algorithms presented in Table 1 have a diminishing-return structure, except for the distorted greedy algorithm.

## F. Faster Implementations of Algorithm 1 and Algorithm 2

We present the algorithmic descriptions of faster implementations of Algorithm 1 and Algorithm 2 when the scoring rule of the meta-algorithm has a diminishing-return structure.

---

**Algorithm 10** A faster implementation of Algorithm 1 when $G$ has a diminishing-return structure

---

**Data:** A set of seller $\mathcal{N}$, a cost profile $\boldsymbol{c}$ from sellers, and a random seed $r$
**Result:** A subset of sellers to purchase services from
$S_0 = \emptyset$   Initialize an empty descending-order priority queue $Q$   **for** $i$ *from* 1 *to* $n$ **do**
| Insert an element to $Q$ with key $i$ and value $G(i, \emptyset, \boldsymbol{c}, 0, r)$
**end**
**while** $k < n$ **do**

> Pop the highest score candidate $i^*$ from $Q$   **while** *True* **do**
>
> > Let the highest score from $Q$ be $s^*$ (without popping the candidate)   **if** $G(i^*, S_k, \boldsymbol{c}, k+1, r) > \max(0, s^*)$ *or*
> > $s^* < 0$ **then**
> > | Break
> > **end**
> > **else**
> > > Pop the highest score candidate $j^*$ from $Q$   Insert an element to $Q$ with key $i^*$ and value $G(i^*, S_k, \boldsymbol{c}, k+1, r)$
> > > $i^* = j^*$
> > **end**
>
> **end**
> **if** $G(i^*, S_k, \boldsymbol{c}, k+1, r) > 0$ **then**
> | $S_{k+1} = S_k \cup \{i^*\}$  $k = k+1$
> **end**
> **else**
> | break
> **end**

**end**
**return** $S_k$

---

---

**Algorithm 11** A faster implementation of Algorithm 2 when $G$ has a diminishing-return structure

---

**Data:** A set of sellers $\mathcal{N}$, a bid profile $\boldsymbol{b}$ from sellers, and a meta algorithm $\mathcal{A}$

**Result:** A subset of sellers to purchase from and a vector of payment to sellers

Generate a random seed $r$ if needed or set $r = 0$ $S^* = \mathcal{A}(\mathcal{N}, \boldsymbol{b}, r)$ computed using Algorithm 10 and record the intermediate
solutions $\{S_0, S_1, \cdots, S_K\}$ **for** $k$ *from* $1$ *to* $K$ **do**

> $i = S_k \setminus S_{k-1}$, $T_0 = S_{k-1}$, and $j = 0$ Initialize an empty descending-order priority queue $Q$ **for** $i \in \mathcal{N} \setminus S_k$ **do**
>
> > | Insert an element to $Q$ with key $i$ and value $G(i, S_{k-1}, \boldsymbol{c}, 0, r)$
>
> **end**
>
> $p_i = b_i$ **while** $j < n - k$ **do**
>
> > Pop the highest score candidate $\ell^*$ from $Q$ **while** *True* **do**
> >
> > > Let the highest score from $Q$ be $s^*$ (without popping the candidate) **if** $G(\ell^*, T_j, \boldsymbol{c}, j + k, r) > \max(0, s^*)$ *or*
> > > $s^* < 0$ **then**
> > >
> > > > | Break
> > >
> > > **end**
> > >
> > > **else**
> > >
> > > > Pop the highest score candidate $\ell'$ from $Q$ Insert an element to $Q$ with key $\ell^*$ and value $G(\ell^*, T_j, \boldsymbol{b}, j + k, r)$
> > > > $\ell^* = \ell'$
> > >
> > > **end**
> >
> > **end**
> >
> > **if** $G(\ell^*, T_j, \boldsymbol{c}, j + k, r) > 0$ **then**
> >
> > > $p_i = \max\left(p_i, \sup\left\{z \geq 0 \mid G\big(i, T_j, (z, \boldsymbol{b}_{-i}), j + k, r\big) > G(\ell^*, T_j, \boldsymbol{b}, j + k, r)\right\}\right)$ $T_{j+1} = T_j \cup \{\ell^*\}$ $j = j + 1$
> >
> > **end**
> >
> > **else**
> >
> > > | break
> >
> > **end**
>
> **end**

**end**

**return** $S^*$ *and* $\boldsymbol{p}$

---

