# OpenReview forum: "Procurement Auctions via Approximately Optimal Submodular Optimization"
_ICML.cc/2025/Conference — ICML 2025 spotlightposter_

### Official Review · Reviewer_PxDv · 2025-03-12

**Overall Recommendation:** 4

**Summary:**

The paper studies the design of procurement auctions with submodular welfare. The problem involves an auctioneer and $n$ sellers, each possessing an item for sale with a private cost $c_i$, representing the minimum price at which they are willing to sell. The auctioneer's valuation over items is given by a monotone submodular function $f$. The mechanism selects a subset $S$ of items based on the sellers' reported costs and determines the payment for each seller. The goal is to design a truthful mechanism that maximizes $f(S) - c(S)$ while ensuring that the total payment does not exceed $f(S)$.

The main contribution of the paper is establishing useful frameworks that transform a reasonable greedy-like submodular optimization algorithm into a truthful mechanism without losing its approximation/competitive ratio in both offline and online settings.  More specifically, the submodular optimization algorithm can determine the item selection strategy for the auctioneer. Then, the authors design corresponding payment rules to ensure truthfulness. The paper further extends these results to the setting of descending auctions. Finally, the proposed mechanisms are empirically evaluated.

## update after rebuttal
I appreciate the authors' rebuttal and will keep my original score.

**Claims And Evidence:**

Yes

**Essential References Not Discussed:**

No

**Experimental Designs Or Analyses:**

Yes, the paper evaluates the proposed mechanisms on a real-world dataset (although the scenario differs slightly from actual applications, it is still reasonable).

**Methods And Evaluation Criteria:**

Yes, the proposed mechanisms are evaluated both theoretically and empirically.

**Other Comments Or Suggestions:**

Section 4 discusses several reasonable assumptions for the meta-algorithm. It might be more readable to assign a name to each assumption.

**Other Strengths And Weaknesses:**

The structure of Section 4 is a little bit weird. It first presents the mechanism framework for the offline setting, followed by Section 4.1, which covers the online mechanism framework. It might be better to move the offline results to Section 4.1, the online results to Section 4.2, or alternatively, move the online results to the appendix.

**Questions For Authors:**

.

**Relation To Broader Scientific Literature:**

The paper makes contributions to mechanism design with a submodular welfare objective. The proposed payment rule computation may influence future work in this area.

**Theoretical Claims:**

Yes, I reviewed the truthfulness proof of the mechanism.

---

> ### Author Rebuttal · Authors · 2025-04-01
>
> We would like to thank the reviewer for taking the time to read our paper and for their valuable feedback.  We will re-organize section 4 and name the assumptions we use for the submodular optimization algorithm.

---

### Official Review · Reviewer_KBrU · 2025-03-13

**Overall Recommendation:** 3

**Summary:**

This paper focuses on procurement auctions where an auctioneer aims to acquire services from strategic sellers with private costs. The quality of services is represented by a submodular function, and the goal is to design efficient mechanisms that maximize the difference between service quality and total seller costs while meeting IC, IR, NAS constraints.

The authors first review existing research on procurement auctions and regularized submodular maximization. Then they show that for the distorted greedy algorithm, a stronger guarantee holds (stronger than previous results). In the mechanism design aspect, they first show that VCG mechanisms satisfy IC, IR, and NAS but are computationally prohibitive. They then develop a framework that can convert all submodular optimization algorithms into sealed-bid mechanisms that meet the desired properties and preserve approximation guarantees. This framework is also extended to the online setting. Additionally, they establish a connection between online submodular optimization and descending auctions, and prove that in the adversarial setting, a descending auction with an exact demand oracle may return a poor solution, but one based on the cost-scaled greedy algorithm can achieve a good approximation guarantee. Finally, The experimental results show that VCG and descending auctions with optimal oracle have high complexity, while greedy-based algorithms have polynomial complexity. In terms of welfare, the direct implementations of approximation algorithms outperform their descending auction variants.

**Claims And Evidence:**

Yes.

**Essential References Not Discussed:**

No, to my knowledge.

**Experimental Designs Or Analyses:**

Yes. No issues have been found so far.

**Methods And Evaluation Criteria:**

Yes.

**Other Comments Or Suggestions:**

I hope the author can analyze the complexity of the algorithm in more detail.

**Other Strengths And Weaknesses:**

No.

**Questions For Authors:**

See suggestions above.

**Relation To Broader Scientific Literature:**

This paper comprehensively studies procurement auctions from multiple aspects.
This paper improves the analysis of the distorted greedy algorithm, advancing regularized submodular optimization.
This paper develops frameworks to transform submodular optimization algorithms into mechanisms for procurement auctions that satisfy IC, IR and NAS, which is related to the literature on mechanism design in procurement auctions.
This paper contributes to the understanding of descending auctions, and makes a bridge between two areas via the reduction from online submodular optimization to descending auctions.

**Theoretical Claims:**

Yes. No issues have been found so far.

---

> ### Author Rebuttal · Authors · 2025-04-01
>
> We would like to thank the reviewer for taking the time to read our paper and for their valuable feedback.
>
> >I hope the author can analyze the complexity of the algorithm in more detail.
>
> Thanks for the comment, notice that all of our algorithms run in polynomial time. For example, in algorithm 2, we need to make at most $O(n)$ many calls to the optimization algorithm in line 284, i.e., one call for each chosen seller. Moreover, for each inner loop starting in line 286, we need to make $O(n \log |B|)$ calls to the scoring function where $|B|$ is the number of possible bids. To summarize, Algorithm 2 makes $O(n)$ calls to the optimization algorithm and $O(n^2 \log |B|)$ calls to the scoring function. We will elaborate on this in the next version of our work. Moreover, our experiments show that they can be implemented even in practical applications. In the next version of our work we will explicitly state the number of oracle calls to the submodular optimization algorithm as well as the extra complexity of our operations, for each of the algorithms we use.

---

### Official Review · Reviewer_fVpD · 2025-03-14

**Overall Recommendation:** 4

**Summary:**

In this paper, they develop a framework to convert a family of greedy algorithms for submodular maximization to a mechanism for procurement actions.
Moreover, they provide an improved analysis of the Distorted Greedy algorithm.
Finally, they consider the case of Descending auctions where they design a mechanism based on an online greedy algorithm for submodular maximization.

**Claims And Evidence:**

All claims are supported by proofs.

**Essential References Not Discussed:**

N/A

**Experimental Designs Or Analyses:**

The experimental design is clear and sound.

Information on the specs of the machine that the experiments where run is missing.
Also, the MIP solver that was used is not mentioned.

**Methods And Evaluation Criteria:**

The proposed algorithms and evaluation criteria make sense.

**Other Comments Or Suggestions:**

- In the description of the Distorted Greedy algorithm (lines 174-180) I would suggest using another symbol instead of $k$ in order to avoid confusion with the cardinality constraint.
The same holds for Stochastic Distorted Greedy.

- I would suggest adding an appendix briefly describing the VCG mechanism for non expert readers.

- There might be a typo in the definition of $u_i^M(b)$ (line 161). Please check.

- I think there might be a typo in line 263. Is it $S_k$ or $S_{k-1}$? In contrast, in Algorithm 2 line 287 you have $S_{k-1}$.

- Algorithm 2 is difficult to follow. Especially, the for loop in lines 286-290. The definition of $p_i$ in the appendix (proof of theorem 4.3) is much clearer.
Also, the variable $i$ is used in the for loop as well as inside the $\max$ (line 289). Is this the same variable? Again the notation in the appendix is much clearer.

- In line 175 (second bullet), I think it should be $G(l_1^\star, \emptyset, ...) > 0$ instead of $G(l, \emptyset, ...)$

**Other Strengths And Weaknesses:**

N/A

**Questions For Authors:**

- Are you familiar with any other work converting greedy algorithms to auction mechanisms? Is this a classic approach?
- Can you clarify what is the complexity of Algorithm 2? I expected that the algorithm would be impractical for medium size inputs however in the experiments it seems to perform well.
- Can you formally define $OPT(b)$ in Appendix C?
- When using the lazy greedy variants is there a significant reduction in running time? Did you run the experiments using the lazy variants?
- Are you going to release the code for the experiments?
- Can you describe potential future research directions?

**Relation To Broader Scientific Literature:**

The paper contains a comprehensive review of submodular maximization algorithms.
The main idea is to transform greedy algorithms for submodular maximization to a mechanism for procurement auctions thus connecting the domain of submodular maximization to that of algorithmic game theory and mechanism design.

**Theoretical Claims:**

I checked the correctness of proposition 4.1 (Appendix C) and theorem 4.3 (Appendix C). I didn't find any errors.

---

> ### Author Rebuttal · Authors · 2025-04-01
>
> We would like to thank the reviewer for taking the time to read our paper and for their valuable feedback. We respond to each of the points they raised below.
>
> >Information on the specs of the machine and the MIP solver.
>
> Thanks for the suggestion, we will add more details about the specs of the machines and the MIP solver in the revision.
>
> >In the description of the Distorted Greedy algorithm (lines 174-180) ...
>
> Thanks for the suggestion, we will make the edit.
>
> >VCG in the appendix.
>
> This is a valid point, we will do that.
>
> >Typo in the definition of $u^M_i(b)$  (line 161).
>
> We believe that this expression is correct; the seller could potentially get paid even if their service isn’t purchased, but it only incurs a cost if it has to provide the service. However, all the mechanisms discussed in this paper satisfies the property that the payment of a seller would be 0 if their service isn’t purchased. Please let us know if this isn’t clear.
>
> >I think there might be a typo in line 263.
>
> Thanks for catching that, this is indeed a typo! It should be $S_{k-1}$.
>
> >Algorithm 2 is difficult to follow.. the variable $i$ is used in the for loop as well as inside the $\max$  (line 289).
>
> We will modify the algorithm, importing notation from the appendix to make it easier to follow. To answer your question, $i$ is the same variable as in the for loop, and we compute the appropriate threshold payments for seller $i$ in line 289 with respect to $S_{k-1}$ and take the max with the current p_i.
>
> >In line 175 (second bullet), I think it should be $G(l_1^*,\emptyset,\ldots) > 0$  instead of  $G(l,\emptyset,\ldots) > 0$.
>
> Thanks for catching that, this is indeed a typo in line 1175.
>
> >Are you familiar with any other work converting greedy algorithms to auction mechanisms? Is this a classic approach?
>
> Converting algorithms to mechanisms is in general a classical approach in algorithmic game theory. For instance, VCG transforms an (optimal) algorithm to a welfare-optimal mechanism. If the optimal algorithm for the underlying problem happens to be greedy, then VCG can be viewed as some such transformation. There have been other works studying black-box transformations from algorithms to mechanisms, under various objectives such as welfare or revenue. Some classical works include Lehmann, Lehmann, and Nisan (2001), Archer and Tardos (2001), Mu'alem and Nisan (2002), Babaioff, Lavi, and Pavlov (2009), Dobzinski and Nisan (2010).
>
> However, most of these settings handle auctions where the designer is *selling* items, and the conversion may not always work if the algorithm does not give optimal outcome. In our setting, we consider converting approximately optimal algorithms and making sure that the conversion creates a mechanism that satisfies the NAS property adds an additional difficulty. For instance, it was not clear to us whether VCG would satisfy that or not. We will discuss some of these works and the differences to our setting in the next version of our work.
>
> >Can you clarify what is the complexity of Algorithm 2?
>
> In algorithm 2, we need to make at most $O(n)$ many calls to the optimization algorithm in line 284, i.e., one call for each chosen seller. Moreover, for each inner loop starting in line 286, we need to make $O(n \log |B|)$ calls to the scoring function where $|B|$ is the number of possible bids. To summarize, Algorithm 2 makes $O(n)$ calls to the optimization algorithm and $O(n^2 \log |B|)$ calls to the scoring function. We will elaborate on this in the next version of our work.
>
> >Can you formally define $OPT(b)$  in Appendix C?
>
> $OPT(b)$ is the *optimal solution* to the optimization problem when sellers report bids $b$, i.e., $OPT(b) \in \argmax_{S \in 2^N} f(S) - \sum_{i \in S} b_i$. We will clarify that in Appendix C.
>
> >When using the lazy greedy variants is there a significant reduction in running time? Did you run the experiments using the lazy variants?
>
> Yes, there is a significant reduction and we used the lazy variants for all algorithms, except for the distorted greedy, which doesn’t admit a diminishing return structure in its scoring function so that lazy greedy cannot be applied.
>
> >Are you going to release the code for the experiments?
>
> We will try to release the code.
>
> >Can you describe potential future research directions?
>
> An immediate direction is to close the gap between the $(1/2,1)$ bound of our descending auction and the $(1-1/e,1)$ we get in the sealed-bid auction. Another interesting problem would be to replace the cost in the objective of the mechanism designer with the payment, i.e., to study an objective of maximizing the surplus of the mechanism designer. It would also be interesting to see if the descending auction can get the same performance as VCG, when we disregard computational considerations. We will elaborate in the next version.

---

### Official Review · Reviewer_zLfD · 2025-03-16

**Overall Recommendation:** 4

**Summary:**

The paper studies a procurement mechanism with objective function f(S) - \sum_{i \in S} p(i) that is truthful, individually rational, and has nonnegative surplus, and provide a bi-criteria approximation.
To the best of my knowledge, this is the first to study such objective function in the procurement auction literature, largely different from the standard procurement auction setting to minimize cost given a utility constraint or the budget-feasible mechanism design setting to maximize utility given an ex-post budget constraint.
The authors provide an improved analysis for the standard Distorted Greedy algorithm for this sake, and design a mechanism based on it.
The authors then extend their results to online mechanism and descending auction.
They further validate their findings through experiments.

**Claims And Evidence:**

All the claims sound clear and well-explained.
One minor issue I found is that the improved analysis of the Distorted Greedy seems a bit orthogonal to the paper's main contribution, though I appreciate the result itself.
One direction might be to make the paper more focused on the mechanism design part, while deferring the improved analysis part to the appendix, but I don't think this to be crucial component to judge the paper's contribution.

**Essential References Not Discussed:**

I don't find any

**Experimental Designs Or Analyses:**

I found no issue

**Methods And Evaluation Criteria:**

Yes

**Other Comments Or Suggestions:**

Figures are a little hard to parse - legends and titles/axes are not very readable until I enlarge my screen enough.

**Other Strengths And Weaknesses:**

The paper is generally well written, and I don't find specific weakness of the paper.
One minor point I'd say is that the paper contains too many results, which are not equally interesting.
The authors might want to more strategically present a part of the results they think will be the most significant.

**Questions For Authors:**

Nothing specific.

**Relation To Broader Scientific Literature:**

The paper's topic is significant to mechanism design and submodular maximization.

**Theoretical Claims:**

I haven't read all the proofs line by line, but the ideas sound reasonable.

---

> ### Author Rebuttal · Authors · 2025-04-01
>
> We would like to thank the reviewer for taking the time to read our paper and for their valuable feedback. We agree with the suggestions and we will do the appropriate reorganization of our content based on their feedback. If our manuscript gets accepted, we will also make sure to utilize the extra space of the camera ready version to enlarge the figures.

---

### Decision · Program_Chairs · 2025-05-01

**Decision:**

Accept (spotlight poster)

**Comment:**

This is a strong and broadly-scoped paper addressing a classic problem, mechanism design for procurement auctions, in light of some new results in submodular optimizations.  All reviewers appreciated the exposition, the depth of theoretical results, and the scope of the setting(s) considered.  This AC would encourage the authors to make the edits proposed in response to fVpD's detailed reviews.